# MalTree: Tracing Malware Evolution from Embeddings at Scale

**Akash Amalan** [1]   **Georgios Smaragdakis** [1]   **Tom J. Viering** [1]

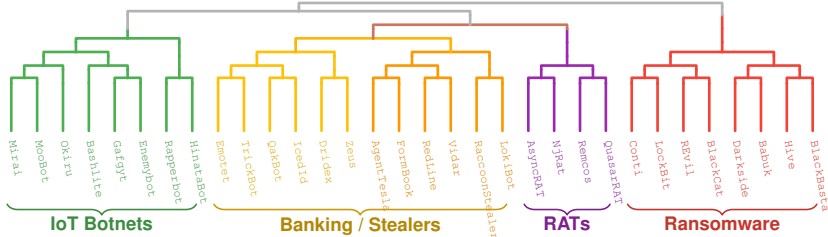

*Figure 1.* Simplified version of phylogenetic tree that illustrates malware evolutionary relationships.

## Abstract

Malware detection remains largely reactive: machine learning models trained on known samples degrade as threats evolve. Understanding evolutionary relationships among malware families can inform proactive defense, but traditional reverse engineering can take months to years to uncover such lineage relationships. We propose MalTree, a framework that applies bioinformatics-inspired phylogenetic techniques (UPGMA and Neighbor-Joining) at scale to model malware evolution automatically using structural, behavioral, and image-based features. We introduce temporal validation using VirusTotal timestamps to assess whether inferred trees reflect actual evolutionary order. MalTree achieves 87% temporal consistency, indicating that inferred evolutionary relationships closely align with real-world emergence timelines. Our analysis shows that some families mutate over 10 times faster than others, suggesting that detection strategies should be tailored to family-specific evolutionary tempos. Case studies, including the Mirai botnet, confirm that inferred relationships from our phylogenetic tree align with documented threat intelligence. Our framework provides a foundation for shifting malware analysis from sample-by-sample classification toward lineage-aware evolutionary modeling.

## 1. Introduction

The battle between malware operators and defenders has long resembled a relentless cat-and-mouse game. Each defensive advance in static or dynamic analysis provokes an adversarial countermeasure: encryption, packing, obfuscation, or metamorphic mutation. As detection algorithms become more sophisticated, malware correspondingly evolves, employing polymorphism to alter its signature or metamorphism to recompile its structure without changing its functionality (Sharma & Sahay, 2014; Badhwar, 2021; Walenstein et al., 2007). This evolutionary contest continues to escalate with the advent of large language models capable of generating novel, functional, and evasive code (Gupta et al., 2023; Shimony & Tsarfati, 2023). Underground variants such as WormGPT explicitly target malware development (Erzberger, 2023). The result is an adaptive ecosystem in which detection must not only recognize existing threats but anticipate their future forms.

**The limits of reactive detection.** Current detection paradigms, whether signature-based, feature-driven, or deep learning-based, remain fundamentally *reactive* (Anderson & Roth, 2018; Schultz et al., 2001; Nataraj et al., 2011). As threats evolve, models trained on past data gradually lose relevance, leading to performance decay (Pendlebury et al., 2019; Jordaney et al., 2017). This sample-centric view overlooks a critical reality: malware variants rarely emerge in isolation. They inherit code from predecessors, share builder toolkits, and evolve through iterative modification, forming *lineages* shaped by code inheritance and adaptation. The continuous interplay between detection and evasion no longer resembles a static taxonomy of threats but rather a co-evolving digital ecosystem, where each defensive innovation exerts selective pressure that accelerates adversarial adaptation, a process parallel to biological evolution.

[1]Delft University of Technology, Delft, The Netherlands. Correspondence to: Akash Amalan <akashamalan53@gmail.com>.

*Proceedings of the 43rd International Conference on Machine Learning*, Seoul, South Korea. PMLR 306, 2026. Copyright 2026 by the author(s).

**Phylogenetics as an evolutionary lens.** To escape the re-active cycle, defenders must transition from recognizing individual samples to modeling how malware families evolve. Phylogenetic trees, foundational in reconstructing ancestral relationships among species and viruses, infer lineage structure from measurable divergence (Saitou & Nei, 1987; Hadfield et al., 2018; Duchene et al., 2020). The same principles of mutation, selection, and inheritance that govern viral evolution also shape malicious software through code reuse and modular propagation (Vinod et al., 2012; Karim et al., 2005). By quantifying evolutionary distances between variants, phylogenetic analysis enables proactive identification of emerging lineages before they proliferate into widespread threats.

**The gap in existing approaches.** Despite this natural analogy, phylogenetic methods remain underexplored in malware analysis. Prior work has applied tree-based clustering to small sample sets using single feature modalities (Karim et al., 2005), or employed minimum spanning trees as phylogenetic proxies without temporal validation (Cozzi et al., 2020). More recent efforts improve algorithmic efficiency but do not analyze evolutionary relationships (He et al., 2023). Crucially, none of these approaches validate whether inferred trees reflect actual emergence timelines, leaving open the question of whether phylogenetic structure captures malware evolution or merely feature similarity.

**Contributions.** In this paper, we propose **MalTree**, a framework for building and validating large-scale phylogenetic trees from malware samples. MalTree extracts multi-modal embeddings, transforms them into distance matrices, and constructs phylogenetic trees using Neighbor-Joining and UPGMA. Our main contributions are:

1. **Large-scale phylogenetic analysis:** We construct validated phylogenetic trees from 103,883 malware samples spanning 538 families in just 11 hours (UPGMA) or 3 days (NJ) on 20 cores, which to our knowledge is the largest such analysis to date. Figure 1 shows 32 representative families.

2. **Temporal validation framework:** We introduce time divergence analysis using VirusTotal (VirusTotal, 2012) timestamps to validate evolutionary directionality, a challenge unexplored in prior work.

3. **Interpretable evolutionary insights:** We develop tree simplification and inter-family analysis methods that reveal relationships (e.g., Mirai lineage) consistent with public cybersecurity intelligence.

4. We release the code and embeddings at: `https://github.com/AJ730/MalwareEvolution`

The remainder of this paper is organized as follows. Section 2 reviews related work. Section 3 introduces phylogenetic preliminaries. Section 4 details the MalTree framework. Section 5 presents experimental results, and Section 6 discusses limitations and future directions.

## 2. Related Work

**Malware Representation Learning.** Image-based approaches convert binaries to visual representations amenable to CNN-based classification (Nataraj et al., 2011; Kalash et al., 2018). Following established methods, we convert malware executables to RGB images using the bin2png (Sultanik, 2020) approach and extract embeddings using a ResNet-50 architecture pre-trained on ImageNet. While we use these CNN-based representations, byte- and image-based features have known limitations: they are sensitive to packing and compilation, with reported accuracies sometimes reflecting dataset or labeling artifacts rather than semantics (Raff et al., 2018). We mitigate this in two ways. First, we compile the malware images from memory dumps instead of the raw binaries. We also fuse the image modality with pseudo-static and dynamic features. Our main contribution lies in what we do with these embeddings: constructing and validating phylogenetic trees at unprecedented scale.

**Evolutionary Analysis of Malware.** Several works have explored evolutionary perspectives on malware. Karim et al. (2005) introduced phylogenetic concepts using sequence alignment, Cozzi et al. (2020) analyzed IoT malware through binary diffing and graph-based clustering, He et al. (2023) addressed computational bottlenecks in tree construction, and Suarez-Tangil et al. (2014) applied hierarchical clustering to Android malware. Our work extends this line at larger scale, building rooted phylogenetic trees rather than proxies and adding temporal validation. Others infer lineage directly from source code: Li et al. (2025) recover genealogies via function-level code-reuse detection, which avoids disassembly noise but is limited to 6,032 GitHub-sourced Windows specimens. We instead operate on 103,883 deployed binaries spanning PE, ELF, and DOS, letting us recover lineages such as the Mirai, Bashlite, and Gafgyt IoT botnets that are absent from source-level corpora.

**Tree Construction Methods.** Phylogenetic tree construction methods fall into two categories: character-based approaches such as Maximum Parsimony and Maximum Likelihood that operate on discrete sequence data, and distance-based methods that construct trees from dissimilarity matrices (Felsenstein, 2004). Character-based methods require sequence alignment, which scales poorly beyond thousands of samples and introduces complexity when converting malware into suitable formats (Izquierdo-Carrasco et al., 2011). Distance-based methods, by contrast, operate directly on

pairwise distances, making them well-suited for continuous embeddings at scale. We employ two such methods: UPGMA (Sokal & Michener, 1958), which assumes constant evolutionary rates, and Neighbor-Joining (Saitou & Nei, 1987), which accommodates rate heterogeneity. Their compatibility with embedding vectors, combined with near-quadratic scaling via RapidNJ (Simonsen et al., 2008), makes them suitable for large-scale malware analysis.

**Tree Validation in Computational Biology.** Constructing a phylogenetic tree is only the first step; validating its accuracy is equally important. In bioinformatics, bootstrap resampling (Felsenstein, 1985) assesses confidence in inferred groupings by resampling columns of genetic sequences, while molecular clock methods (Drummond et al., 2006) calibrate trees using known split times from fossil records. However, these approaches assume either that sequence positions evolve independently or that external timing references exist. Neither assumption holds for embedding-based malware analysis, where we have continuous vectors rather than genetic sequences and no ground-truth divergence dates. To address this gap, we introduce an alternative validation approach using VirusTotal submission timestamps as temporal anchors. These timestamps typically record when samples are first reported by security monitors, providing a reliable proxy for emergence time.

## 3. Preliminaries

This section establishes the mathematical foundations for phylogenetic analysis.

### 3.1. Phylogenetic Trees

**Taxa and Families.** In classical phylogenetics, a *taxon* (plural: *taxa*) denotes a taxonomic unit at any rank—a species, genus, or higher grouping—that serves as the basic object of evolutionary analysis. We adopt this terminology for malware: a **taxon** is a single malware executable. Formally, let $\mathcal{S} = \{s_1, s_2, \ldots, s_n\}$ denote a finite set of $n$ taxa, where each element $s_i \in \mathcal{S}$ represents one distinct binary. Two samples $s_i \neq s_j$ are considered distinct taxa even if they belong to the same malware family.

Let $\mathcal{S}$ denote the set of malware samples that pass our VirusTotal validation process (see figure 7 in Appendix C). A *malware family* $\mathcal{F} \subseteq \mathcal{S}$ is a subset of taxa that share common ancestry and exhibit similar behavioral or structural characteristics. Family membership is determined by antivirus vendor consensus via VirusTotal labels. Each sample belongs to exactly one family, partitioning $\mathcal{S}$ into $K$ disjoint families $\{\mathcal{F}_1, \mathcal{F}_2, \ldots, \mathcal{F}_K\}$.

**Tree Structure.** A *phylogenetic tree* over $\mathcal{S}$ is a connected acyclic graph $\mathcal{T} = (V, E, w)$ consisting of:

- $V = L \cup I$ is the vertex set, partitioned into *leaves* $L$ and *internal nodes* $I$;

- $E \subseteq V \times V$ is the edge set;

- $w : E \to \mathbb{R}_{\geq 0}$ assigns a non-negative *branch length*.

Each taxon corresponds to a leaf in the tree, so $|L| = n$, and we use $s_i$ to denote both a taxon and its corresponding leaf. Internal nodes represent hypothetical ancestors from which observed taxa diverged; these are not directly observable but inferred from the data. The tree encodes evolutionary history: the root represents the most ancient common ancestor, time progresses downward toward the leaves (extant samples), and edges represent lineage segments along which mutations accumulate. Branch lengths quantify the degree of divergence between connected nodes.

A phylogenetic tree may be *rooted* or *unrooted*. A rooted tree has a designated root node representing the common ancestor of all taxa; edges are implicitly directed from ancestors to descendants, establishing evolutionary directionality (Figure 2). An unrooted tree is an undirected graph that specifies pairwise relationships without indicating which node is ancestral.

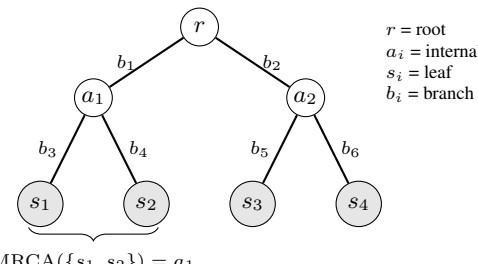

*Figure 2.* A rooted phylogenetic tree with four taxa. Leaves (shaded) correspond to observed samples; internal nodes represent inferred ancestors.

**Path Distance.** For any two nodes $u, v \in V$, let $\mathrm{path}(u, v) \subseteq E$ denote the unique path between them. The *path distance* $d_{\mathcal{T}}(u, v) = \sum_{e \in \mathrm{path}(u,v)} w(e)$ defines a metric on $V$; when both nodes are leaves, this is the *patristic distance*.

**Most Recent Common Ancestor (MRCA).** In a rooted tree, each node $a$ induces a *subtree*; nodes in this subtree are called *descendants* of $a$. For a non-empty subset $S \subseteq L$, the *Most Recent Common Ancestor*, denoted $\mathrm{MRCA}(S)$, is the unique node $a \in V$ satisfying: (i) every leaf in $S$ is a descendant of $a$, and (ii) no child of $a$ satisfies condition (i). For example, in Figure 2, $\mathrm{MRCA}(\{s_1, s_2\}) = a_1$. For a

leaf $s_i$ and an ancestor $a$, we write $L_i = d_{\mathcal{T}}(s_i, a)$ to denote the path distance from the leaf to that ancestor.

**Distance Matrices.** Phylogenetic tree construction requires a measure of dissimilarity between taxa. We represent each taxon $s_i \in \mathcal{S}$ by an embedding vector $\mathbf{e}_i \in \mathbb{R}^d$, obtained via embedding extraction. The distance matrix $\mathbf{D} \in \mathbb{R}^{n \times n}$ is defined as $D_{ij} = \|\mathbf{e}_i - \mathbf{e}_j\|_2$, under the assumption that pairwise embedding distances approximate evolutionary divergence.

### 3.2. Tree Construction Algorithms

We employ two distance-based methods suitable for embeddings at scale.

**UPGMA.** The Unweighted Pair Group Method with Arithmetic Mean (Sokal & Michener, 1958) iteratively merges the two closest clusters until a single tree remains, assuming a *molecular clock* (constant evolutionary rate across lineages). Initializing each taxon as its own cluster $\mathcal{C} = \{\{s_1\}, \{s_2\}, \ldots, \{s_n\}\}$, each step:

1. Finds the pair $(C_i, C_j)$ minimizing average inter-cluster distance:

$$d(C_i, C_j) = \frac{1}{|C_i| \cdot |C_j|} \sum_{s_p \in C_i} \sum_{s_q \in C_j} D_{pq} \quad (1)$$

2. Merges $C_i$ and $C_j$ into $C_{ij}$, creating an internal node at height $h = d(C_i, C_j)/2$.

3. Updates $\mathcal{C} \leftarrow (\mathcal{C} \setminus \{C_i, C_j\}) \cup \{C_{ij}\}$ and recomputes distances.

UPGMA produces a *rooted ultrametric tree* in $O(n^2)$ time (Day & Edelsbrunner, 1984).

**Neighbor-Joining.** NJ (Saitou & Nei, 1987) relaxes the molecular clock, allowing lineages to evolve at different rates. Starting from a star topology, NJ iteratively joins pairs minimizing total branch length:

1. Compute the $Q$-matrix, where lower $Q_{ij}$ indicates that $i$ and $j$ are neighbors:

$$Q_{ij} = (r - 2) D_{ij} - \sum_{k=1}^{r} D_{ik} - \sum_{k=1}^{r} D_{jk} \quad (2)$$

where $r$ is the number of remaining taxa.

2. Select the pair $(i, j)$ minimizing $Q_{ij}$ and join them via new internal node $u$.

3. Assign branch lengths from $u$:

$$b_{iu} = \frac{1}{2} D_{ij} + \frac{1}{2(r - 2)} \left( \sum_{k=1}^{r} D_{ik} - \sum_{k=1}^{r} D_{jk} \right) \quad (3)$$
$$b_{ju} = D_{ij} - b_{iu}$$

4. Update distances: $D_{uk} = (D_{ik} + D_{jk} - D_{ij})/2$ for all remaining $k$.

Standard NJ has $O(n^3)$ complexity; we use RapidNJ (Simonsen et al., 2008) for near-$O(n^2)$ performance. Unlike UPGMA, NJ produces an *unrooted tree*, requiring a separate rooting step.

### 3.3. Rooting Methods

We consider two approaches for placing a root.

**Outgroup Rooting.** If a taxon $s_o \in \mathcal{S}$ is known *a priori* to be more distantly related to the remaining taxa than they are to each other, the root is placed on the branch connecting $s_o$ to the rest of the tree (Farris, 1972). In our setting, we select the sample with the earliest VirusTotal first-submission timestamp as the outgroup, under the assumption that earlier submissions approximate ancestral lineages.

**Midpoint Rooting.** The root can be placed at the midpoint of the longest path in the tree. Formally, let

$$(s^*, s^{**}) = \underset{(s_i, s_j) \in L \times L}{\arg\max} \; d_{\mathcal{T}}(s_i, s_j) \quad (4)$$

be the pair of leaves with maximum patristic distance. The root $r$ is positioned on the path between $s^*$ and $s^{**}$ such that $d_{\mathcal{T}}(r, s^*) = d_{\mathcal{T}}(r, s^{**})$ (Farris, 1972). This method implicitly assumes approximate rate constancy across the tree.

## 4. MalTree Framework

We present MalTree, a framework for constructing and validating large-scale phylogenetic trees from malware samples. Figure 3 illustrates our pipeline: multi-modal embedding extraction, embedding fusion with dimensionality reduction, distance matrix construction, and tree generation.

### 4.1. Multi-Modal Embedding Extraction

We extract three complementary embeddings from each malware sample, capturing structural, behavioral, and visual characteristics. Rather than analyzing raw executables, which are often packed or obfuscated, we perform analysis on memory dumps extracted during controlled execution in virtual machines, revealing decrypted code and runtime state.

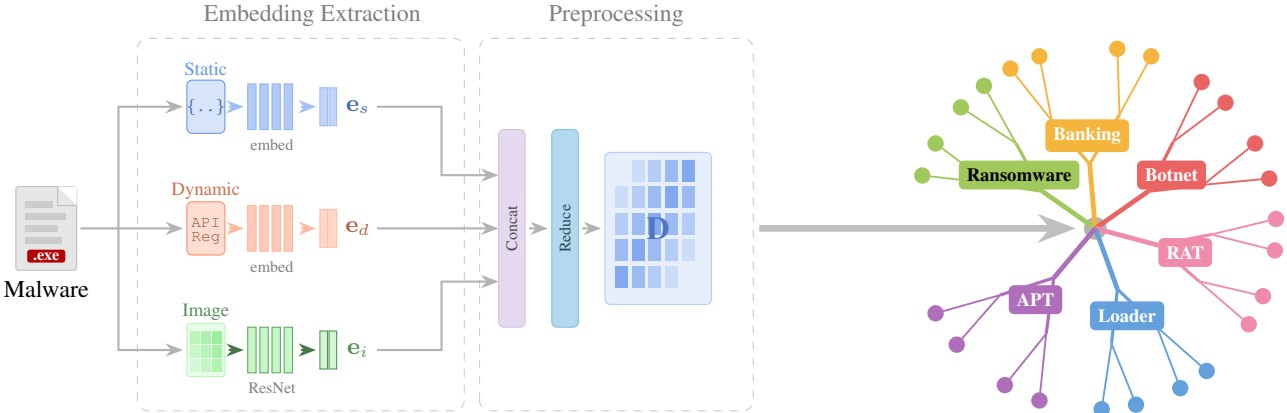

*Figure 3.* **MalTree pipeline.** *Left:* Multi-modal embedding extraction produces pseudo-static ($\mathbf{e}_s$), dynamic ($\mathbf{e}_d$), and image ($\mathbf{e}_i$) representations. These are concatenated and reduced, from which pairwise distances yield matrix $\mathbf{D}$. *Right:* Tree construction via Neighbor-Joining reveals family-level structure, with clades corresponding to functional categories.

**Pseudo-static embedding ($\mathbf{e}_s \in \mathbb{R}^{3512}$).** From each memory dump, we use the LIEF library (Thomas, 2017) to extract executable sections by identifying format signatures (MZ for Windows PE/DOS, \x7fELF for Linux). We extract 27 features including byte histograms (256-d), byte entropy histograms (256-d), string tables, header information, section metadata, imports, exports, and opcode sequences. Structured features are serialized to JSON and encoded via OpenAI's `text-embedding-3-large` model (OpenAI, 2024) to produce 3,000-dimensional vectors. The final embedding concatenates histogram features (512-d) with text embeddings (3,000-d), yielding 3,512 dimensions. A comprehensive description of all extracted features is provided in Appendix M.

**Dynamic embedding ($\mathbf{e}_d \in \mathbb{R}^{1000}$).** Behavioral traces are collected from sandbox execution via ANY.RUN (ANY.RUN, 2016) and VirusTotal APIs, capturing 15 feature categories spanning file, command, process, registry, service, network, and mutex activity. Entry limits (30 files, 20 processes, 10 registry keys) ensure tractable feature sizes; the complete category list is provided in Appendix B. These behavioral features are encoded via the same text embedding model with output dimensionality set to 1,000 (empirically chosen; higher dimensions showed no improvement).

**Image embedding ($\mathbf{e}_i \in \mathbb{R}^{2048}$).** Memory dumps are converted to RGB images using the bin2png method (Sultanik, 2020), which maps consecutive three-byte chunks to RGB pixel values. Images are resized to $224 \times 224$ pixels with bilinear interpolation and normalized using standard ImageNet preprocessing (Deng et al., 2009). We employ ResNet-50 (He et al., 2016) with two-stage training: pre-training on public malware image benchmarks, specifically MalImg (Agarap, 2025), MaleVis (Bozkir et al., 2019), and MalNet (Freitas et al., 2022), achieves 85% accuracy, while fine-tuning on our dataset yields approximately 95% family classification accuracy. The resulting 2,048-dimensional penultimate layer activation serves as $\mathbf{e}_i$.

**Embedding Fusion and Dimensionality Reduction.** The embeddings are complementary: $\mathbf{e}_s$ encodes syntactic structure, $\mathbf{e}_d$ captures runtime behavior, and $\mathbf{e}_i$ reflects byte-level patterns. We fuse them via concatenation, normalization, and a two-layer network (cross-entropy loss on family labels), reducing 6,560 dimensions to 1,000 for computational tractability (pairwise distance computation scales quadratically with dimension). After training, we discard the classification head and extract the 1,000-d representation for tree construction. For detailed discussion of the embedding extraction process refer to Appendix L.

### 4.2. Temporal Validation Framework

Standard phylogenetic validation assumes sequence data (bootstrap resampling (Felsenstein, 1985)) or known divergence times (molecular clock calibration (Drummond et al., 2006)). Neither holds for embedding-based malware analysis: we have no genetic sequences, and divergence times are unknown. We therefore validate by comparing tree-inferred divergence order against VirusTotal first-submission timestamps as proxies for emergence time, using first-submission rather than file creation dates, as the latter proved unreliable (e.g., samples dated 1970 despite families appearing after 2010).

**Family membership.** Throughout this analysis, family membership is determined by VirusTotal labels. These labels partition samples into families, enabling us to distinguish between intra-family comparisons (samples from the same family) and inter-family comparisons (samples from different families).

**Outlier handling.** Outliers can distort tree topology by shifting MRCAs toward the root. We flag them per family using median lateral distance (leaf-to-leaf distance within the family), marking samples beyond $1.5 \times \mathrm{IQR}$ above the family median. We report temporal consistency on the full tree and treat outlier removal only as a robustness check: as Appendix F details, removal slightly raises consistency rather than inflating it, confirming the reported score is not a filtering artifact.

**Inferring divergence order.** Our analysis rests on a path-length assumption applied to siblings sharing an immediate parent: if $L_i = d_{\mathcal{T}}(s_i, a) < L_j$ for their shared ancestor $a$, then $s_i$ emerged earlier. This approach requires only local consistency within sibling pairs rather than constant mutation rates across all lineages (see Appendix G). Figure 4 illustrates two cases:

**Case (i) Intra-family:** For two samples from the same family, we identify their shared MRCA and compare path lengths. If $L_i < L_j$, we infer $t(s_i) < t(s_j)$, i.e., sample $s_i$ emerged before $s_j$. In Figure 4, samples $A_1$ and $A_4$ share ancestor $a_A$; comparing $L_1$ and $L_4$ determines their relative emergence order.

**Case (ii) Inter-family:** For samples from different families (e.g., $A_1$ and $B_1$ in Figure 4), the shared ancestor is typically a deep internal node (here, root $r$). The long evolutionary paths to deep nodes introduce greater distance estimation error, and individual sample comparisons are noisy due to within-family variation. Instead, we compare families as aggregates: for each family, we compute the median distance from all its leaves to $r$ (denoted $\tilde{d}_A$ and $\tilde{d}_B$), using the median to robustly represent each family's typical divergence depth. If $\tilde{d}_A < \tilde{d}_B$, we infer Family A diverged before Family B.

**Temporal consistency score.** We validate trees by comparing the tree-inferred divergence order with timestamps. Following Felsenstein (1985), we focus on shallow divergences where relationships are most reliable: for each leaf, we identify its immediate parent and compare all sibling pairs. A pair $(s_i, s_j)$ is *temporally consistent* if the tree-inferred order matches the timestamp order. The *temporal consistency score* is the proportion of consistent pairs across all such comparisons (Appendix G).

**Embedding drift analysis.** To test whether UPGMA's molecular clock assumption (uniform mutation rates) holds for malware, we measure drift in sample embeddings across years. For each family spanning multiple years, we compute Euclidean distances between embedding vectors of samples from different years. Specifically, for each sample in year $y$, we compute distances to all samples in years $y' > y$, recording the minimum and maximum distances per family

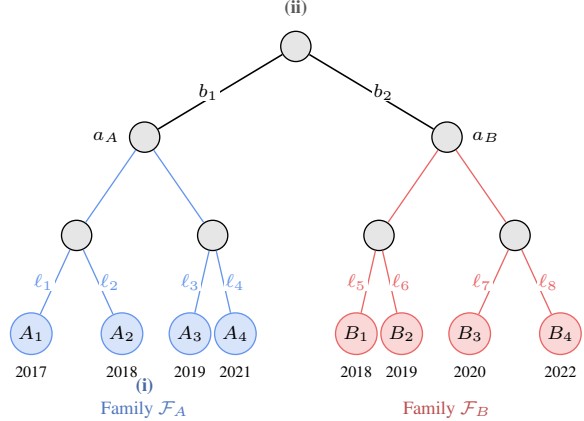

**Cases:** (i) intra-family (compare $L_i$ to shared MRCA); (ii) inter-family (compare median distances to root).

*Figure 4.* Phylogenetic tree for temporal analysis. Leaves show first-submission year. If $L_i < L_j$ for samples sharing an MRCA, then $t(s_i) < t(s_j)$ should hold.

(see Appendix J for details). If mutation rates were uniform, min/max drift values would be consistent across families. High variance indicates non-uniform evolution, favoring NJ over UPGMA.

**Inter-family evolutionary inference.** Beyond validation, we use trees to infer evolutionary relationships between families. For each family pair, we identify their shared MRCA and compute the median path distance from each family's leaves to this ancestor. An edge is directed from the family with shorter median distance (earlier-diverging) to the family with longer median distance (later-diverging), with the edge weight equal to the source family's median distance to the MRCA. To focus on primary lineages, we simplify the graph by retaining only the minimum-weight outgoing edge from each node. Lower edge weights thus indicate closer phylogenetic relationships, while higher weights suggest weaker support. Full details are provided in Appendix H.

## 5. Experiments

We evaluate MalTree on three fronts: whether multimodal embeddings capture family-discriminative structure, whether constructed trees reflect temporal evolution, and whether inferred relationships align with documented threat intelligence.

### 5.1. Experimental Setup

**Dataset.** We curate 103,883 malware samples spanning 538 families from public repositories (Malware-Bazaar (abuse.ch, 2020), vx-underground (vx-underground, 2019), VirusTotal (VirusTotal, 2012)) and additional sources under non-disclosure agreements. The family labels are

verified via VirusTotal consensus (Appendix C). Samples include PE, ELF, and DOS executables with timestamps that span 2010–2023; family sizes range from 10 to over 5,000.

**Methods.** We compare two tree construction algorithms: UPGMA and Neighbor-Joining (NJ). For NJ, we evaluated three rooting strategies: outgroup (oldest sample as root), midpoint (root at the longest path center), and default (arbitrary root placement). We use RapidNJ (Simonsen et al., 2008) to achieve near-$O(n^2)$ performance through heuristic neighbor selection and parallelization across 20 cores on the HPC cluster, with trees rooted using ete3 (Huerta-Cepas et al., 2016). Full hardware specifications are reported in Appendix E.

**Metrics.** For embedding quality, we use family classification accuracy with logistic regression as a linear probe; high accuracy indicates linearly separable features suitable for distance-based phylogenetics. For tree quality, we use *temporal consistency*: the proportion of sample pairs where branch-length ordering matches timestamp ordering from VirusTotal metadata.

## 5.2. Embedding Evaluation

Table 1 shows classification accuracy using stratified 10-fold cross-validation. Combined embeddings (93.69%) outperform individual pseudo-static and dynamic embeddings, confirming that fusion preserves complementary information. Image embeddings achieve highest accuracy (94.91%), suggesting visual structure strongly discriminates families.

*Table 1.* Family classification accuracy from embeddings.

| Embedding | Accuracy (%) |
|---|---|
| Image ($\mathbf{e}_i$) | $94.91 \pm 0.16$ |
| Combined ($\mathbf{e}$) | $93.69 \pm 0.19$ |
| Pseudo-static ($\mathbf{e}_s$) | $87.51 \pm 0.48$ |
| Dynamic ($\mathbf{e}_d$) | $87.23 \pm 0.19$ |
| Random guessing | $19.36 \pm 0.17$ |

## 5.3. Tree Validation

Table 2 compares temporal consistency across tree construction and rooting methods. NJ with outgroup rooting achieves the highest consistency (87.1%), confirming that temporal metadata provides effective rooting. UPGMA performs best in its nominal configuration (86.0%) because it inherently produces rooted trees; additional rooting disrupts its structure. Midpoint rooting performs poorly for both methods, indicating the longest-path heuristic does not reflect malware evolution. The 87% consistency validates that embedding distances approximate evolutionary divergence, as random ordering would yield approximately 50%. Notably, VirusTotal timestamps are independent of family

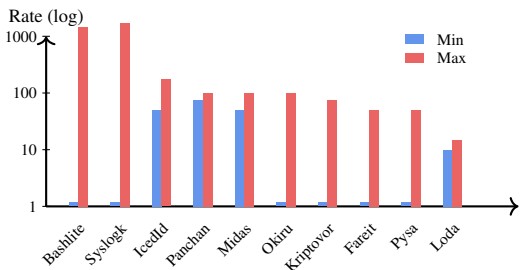

*Figure 5.* Embedding drift (distance/year, log scale) varies substantially across families.

labels used during embedding extraction, reinforcing that embeddings capture genuine evolutionary structure rather than merely reconstructing the training taxonomy. The reported 87.1% is measured on the full tree before any filtering. Removing 5,385 intra-family outliers (of 103,883) raises consistency only to 88.5%, a +1.4 point change rather than the large jump selection bias would produce, confirming the score is not a filtering artifact.

NJ outperforms UPGMA when properly rooted because malware families evolve at non-uniform rates, violating UP-GMA's clock assumption. Figure 5 confirms this: Bashlite and Syslogk exhibit over 10 times higher drift. The three modalities also agree on which families evolve quickly (Appendix O).

*Table 2.* Temporal consistency (year), computed on the full tree of all 103,883 samples before outlier removal. Month-level patterns are similar (Appendix A)

| Method | Default | Outgroup | Midpoint |
|---|---|---|---|
| NJ | 0.811 | **0.871** | 0.631 |
| UPGMA | 0.860 | 0.601 | 0.562 |

## 5.4. Phylogenetic Tree

The resulting NJ tree has 103,883 leaves corresponding to individual malware samples, and 103,882 internal nodes (inferred ancestors). The tree has depth 17, average node degree 2, and maximum node degree 116. For Figure 1, we selected 32 representative families (8 per functional category) with at least 50 samples. A visualization with all families and most confident relations can be viewed here: https://aj730.github.io/PhylogeneticsForMalware/inter_family_vis.html For more methodological detail see Appendix N.

## 5.5. Case Study: Mirai Botnet

To validate inferred relationships against documented threat intelligence, we examine the Mirai botnet, responsible for the 2016 DDoS attack on DNS provider Dyn that dis-

rupted Twitter, Netflix, and Reddit (Antonakakis et al., 2017). Its source code leak spawned numerous variants, making it ideal for validation. Figure 6 shows the inter-family subgraph. MalTree identifies five Mirai descendants (red): **Bashlite** ($w = 9.6$), with confirmed shared code lineage (Akamai Technologies, 2016); **Okiru** ($w = 10.0$), documented by MalwareMustDie as the first Mirai variant targeting ARC processors (MalwareMustDie, 2018); **MooBot** ($w = 14.3$), identified as a Mirai derivative with enhanced DDoS capabilities (The Hacker News, 2022); **Gafgyt** ($w = 17.1$), reported by Trend Micro as sharing RCE exploit techniques with Mirai (Trend Micro Research, 2019); and **RapperBot** ($w = 20.8$), an IoT botnet that borrowed Mirai's SSH brute-forcing code. All five variants share process-management primitives (`fork`, `kill`, `prctl`) and behavioral patterns including `/proc/[pid]/cmdline` probing for anti-analysis evasion.

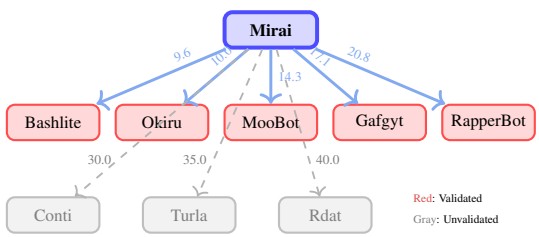

*Figure 6.* Mirai inter-family subgraph with edge weights. Red: validated by threat intelligence; gray: lacking corroborating evidence. Lower weights indicate stronger phylogenetic support.

We corroborate these relationships with static features that are *independent* of the embeddings used to build the tree, computing Jaccard similarity over combined import/export symbol sets extracted with LIEF (Appendix M). Table 3 reports the result within the lineage. Mirai and Okiru overlap strongly (0.82), sharing 44 of their 49–51 unique symbols across networking, process control, memory management, and file I/O; the few Okiru-specific functions (`atoi`, `getdtablesize`, `getuid`, `inet_ntoa`, `sysconf`) are consistent with its documented ARC-architecture adaptation (MalwareMustDie, 2018). The tree places the two as a parent–child pair ($w = 10.0$), and the 82% overlap confirms code reuse. Bashlite shows moderate overlap (0.58), consistent with the lineage reported by Akamai Technologies (2016), and receives the lowest edge weight ($w = 9.6$).

Gafgyt and MooBot show low symbol Jaccard (0.07, 0.04), but this reflects compilation rather than dissimilarity. Gafgyt is statically linked, exporting 1,377 symbols against only 3 imports, while MooBot ships a different C runtime (`nptl`/`__cxa` versus uClibc); in both cases build-specific symbols dominate the union and depress the score. Aggregate comparison therefore misses the relationship, but MooBot's export table still retains Mirai-specific function names such as `attack_-method_udpgeneric`, `attack_udp_ovhhex`,

*Table 3.* Jaccard similarity of combined import/export symbol sets within the Mirai lineage. Low Gafgyt and MooBot values reflect static linking and runtime differences rather than absence of lineage, as the Mirai-specific function names in MooBot's export table confirm.

| | Mirai | Okiru | Bashlite | Gafgyt | MooBot |
|---|---|---|---|---|---|
| Mirai | — | 0.82 | 0.58 | 0.07 | 0.04 |
| Okiru | 0.82 | — | 0.60 | 0.06 | 0.05 |
| Bashlite | 0.58 | 0.60 | — | 0.17 | 0.04 |
| Gafgyt | 0.07 | 0.06 | 0.17 | — | 0.05 |
| MooBot | 0.04 | 0.05 | 0.04 | 0.05 | — |

`setup_connection`, `resolve_cnc_addr`, and `anti_gdb_entry`, providing direct evidence of code inheritance that import-table overlap alone fails to capture.

Beyond pairwise overlap, the symbol tables expose the *internal structure* of the clade, distinguishing kinds of relationship a single similarity score cannot. Mirai exports 37 `attack_*`, `scanner_*`, and `killer_*` functions, and counting how many each family carries verbatim reveals distinct evolutionary modes. MooBot inherits 14 of the 37, the complete `attack_*` framework, and its family-unique functions are *new* attack methods built on it (`attack_tcp_bypass`, `attack_tcp_syndata`, `attack_icmpecho`), the signature of a source fork that extended the codebase. Bashlite instead carries only the `killer_*`/`scanner_*` kill-and-scan functions and none of the attack code, consistent with the documented history in which Mirai's authors borrowed Bashlite's scanning subsystem: a shared module, not a fork. Gafgyt's family-unique functions follow a separate DDoS-for-hire naming convention (`SendOVH`, `NUKE`, `voltstd`), consistent with the tree assigning it a larger edge weight (17.1) than the closer forks. These distinctions, fork versus borrowed subsystem versus parallel lineage, are recoverable only from the code and add functional resolution to the tree's topology.

The graph also suggests connections to Conti ($w = 30.0$), Turla ($w = 35.0$), and Rdat ($w = 40.0$) malware, which lack corroborating evidence and exhibit substantially higher edge weights than the verified relationships, indicating weaker phylogenetic support. Consistently, these families share no Mirai-specific symbols, only generic system imports. This weight separation demonstrates MalTree's utility in both confirming documented relationships and identifying hypotheses requiring additional validation. For additional case studies see Appendix I.

# 6. Discussion

**From Reactive Detection to Evolutionary Modeling.**
The central contribution of this work is demonstrating that phylogenetic analysis, long established in biological do-

mains, can be successfully applied to malware at unprecedented scale. By constructing validated trees from over 103,000 samples across 538 families, we show that the evolutionary lens offers structural insights invisible to sample-by-sample classification. The 87.1% temporal consistency achieved by Neighbor-Joining with outgroup rooting indicates that phylogenetic algorithms such as NJ and UPGMA can transform raw embedding distances into branch lengths that meaningfully reflect evolutionary relationships, providing a foundation for lineage-aware malware analysis.

NJ consistently outperforms UPGMA when properly rooted because malware families exhibit heterogeneous mutation rates. Embedding drift analysis (Figure 5) reveals that families like Bashlite and Syslogk exhibit over 10 times higher maximum embedding drift than other families. This variability violates UPGMA's molecular clock assumption, which requires constant evolution rates across all lineages. The finding has practical implications: detection strategies may need to be tailored to the evolutionary tempo of specific malware families.

**Methodological Refinement.** Several aspects of our methodology warrant discussion. Our path-length assumption (that shorter branch lengths within sibling pairs correspond to earlier emergence) is substantially weaker than a molecular clock assumption, as it does not require constant evolutionary rates. However, it still presumes local consistency in malware development, and may break down for families exhibiting highly irregular or non-linear evolution. Although this assumption achieves 87% temporal consistency in our experiments, this should be further validated. Our analysis also relies on VirusTotal family labels assigned by majority AV consensus, which inherits the known noise in raw AV labels; purpose-built aggregators such as AVclass (Sebastián et al., 2016) and the more recent ClarAVy (Joyce et al., 2025) resolve aliases and generic tokens more reliably and would likely yield cleaner assignments, a natural improvement for future versions of the pipeline. Furthermore, RapidNJ trades exactness for computational tractability, which may also partially distort the tree. Our approach to dealing with outliers also requires further validation. These represent future work for our framework.

**Opportunities for advanced evolutionary modeling.** Phylogenetic trees assume bifurcating evolution, yet malware commonly incorporates code from multiple sources through copy-paste reuse or shared toolkits. Phylogenetic networks, which model reticulate evolution, offer a principled alternative but remain computationally challenging at scale (Huson & Scornavacca, 2010; Moret et al., 2004). Additionally, distance-based algorithms inherently connect all families; establishing principled confidence thresholds for distinguishing meaningful evolutionary relationships from spurious connections remains unsolved. Finally, our current approach requires rebuilding the tree with each new sample. Online phylogenetic algorithms that incrementally update trees could enable real-time deployment (Dinh et al., 2017).

**Implications for Security Practice.** For security practitioners, MalTree complements existing workflows by producing evolutionary insights in weeks rather than the months required for manual reverse engineering. The inter-family analysis provides hypotheses for investigation: confirmed relationships like Mirai-Gafgyt validate known intelligence, while high-weight connections flag candidates for deeper examination. Our case studies reveal that MalTree recovers code evolution, delivery chain associations, and functional convergence, which edge weights alone cannot distinguish. The Conti ecosystem further demonstrates that phylogenetic methods capture functional rather than organizational relationships, as functionally diverse toolsets scatter across unrelated hubs despite shared authorship. Practitioners should cross-reference results with external threat intelligence; MalTree accelerates expert analysis rather than replacing it.

# 7. Conclusion

We presented MalTree, a framework for constructing large-scale phylogenetic trees from malware samples. We achieve 87.1% temporal consistency on 103,883 samples spanning 538 families, and inter-family analysis correctly recovers documented relationships including the Mirai botnet lineage. The consistency between our inferred trees and external validation, both temporal and from threat intelligence, suggests that phylogenetic methods can capture meaningful evolutionary structure in malware. We hope MalTree provides security researchers with a starting point to move beyond classifying individual samples toward understanding how malware families evolve over time.

## Impact Statement

This work advances defensive cybersecurity by enabling automated evolutionary analysis of malware. While the methodology could theoretically be reproduced by adversaries using publicly available samples, the evolutionary relationships we uncover are largely documented in existing threat intelligence and are already known to malware authors through their own development practices. The asymmetric benefit favors defenders, who lack the insider knowledge that adversaries possess about their own codebases. We release embeddings rather than samples, and our framework provides no capability to generate new malware.

## Acknowledgements

The authors thank VirusTotal and ANY.RUN for providing research access to their datasets and tools, and Harm Griffioen for his valuable feedback and comments. Research reported in this work was facilitated by computational resources and support of the Delft AI Cluster (DAIC) at TU Delft. This work was funded by the European Union under the Horizon Europe Programme as part of projects Safe-Horizon (#101168562) and RECITALS (#101168490).

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

## A. Additional Temporal Validation Results

Month-level temporal consistency confirms year-level findings: NJ with outgroup rooting achieves the highest consistency (0.853), while UPGMA performance degrades further at finer temporal resolution.

*Table 4.* Temporal consistency (month).

| Method | Nominal | Outgroup | Midpoint |
|--------|---------|----------|----------|
| NJ | 0.793 | **0.853** | 0.597 |
| UPGMA | 0.569 | 0.553 | 0.507 |

## B. Dynamic Feature Categories

Table 5 lists the 15 behavioral feature categories collected per sample from ANY.RUN (ANY.RUN, 2016) and VirusTotal APIs. Entry limits cap the number of items recorded per category to keep feature sizes tractable (30 files per file-operation category, 20 processes, 10 registry keys).

*Table 5.* Dynamic feature categories collected from sandbox execution.

| Group | Categories |
|-------|-----------|
| File operations | `files_opened`, `files_written`, `files_deleted` |
| Command execution | `commands_executed` |
| Process operations | `processes_created`, `processes_terminated`, `processes_injected` |
| Registry | `registry_keys_created`, `registry_keys_deleted`, `registry_values_modified` |
| Services | `services_created`, `services_started` |
| Network | `dns_queries`, `network_connections` |
| Mutexes | `mutexes_created` |

## C. Malware Sample Validation Process

Figure 7 illustrates our validation process for ensuring sample quality and accurate family labels using VirusTotal consensus.

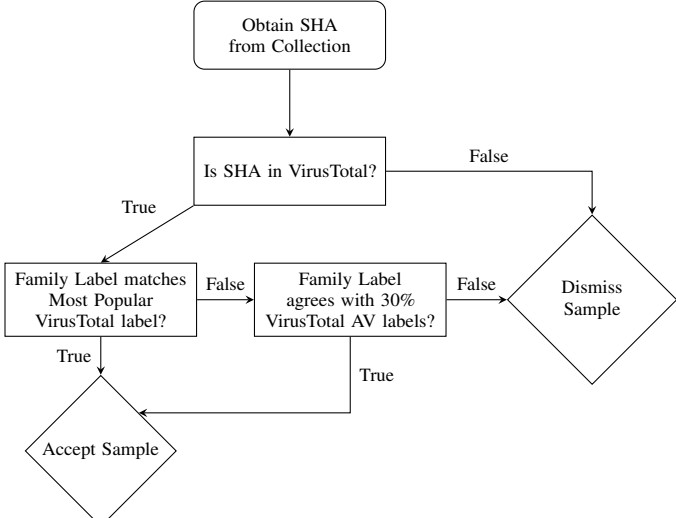

*Figure 7.* Flowchart of how we validate our samples

# D. Linear Probe Training

This appendix provides reproducibility details for the linear-probe evaluation reported in Table 1.

We use multinomial logistic regression as a linear probe to evaluate the family-discriminative content of each embedding. The classifier is implemented in scikit-learn 1.4.1.post1 (Pedregosa et al., 2011) (Python 3.9) as a pipeline of `StandardScaler` (zero-mean, unit-variance per feature) followed by `LogisticRegression`.

**Hyperparameter selection.** Hyperparameters are chosen via nested cross-validation: 10 outer `StratifiedKFold` splits, with model selection performed on an inner 5-fold `GridSearchCV` (scoring on accuracy), refit on the outer training fold, and evaluated on the outer test fold. The search grid is shown in Table 6.

*Table 6.* Hyperparameter grid for the linear-probe logistic regression. The configuration $C{=}0.1$, `penalty=l2`, `solver=lbfgs` was selected in all 10 outer folds.

| Parameter | Values |
|---|---|
| penalty | $\{l1, l2\}$ |
| C | $\{10^{-5}, 10^{-4}, 10^{-3}, 0.1\}$ |
| solver | lbfgs (l2 only), liblinear (l1 and l2) |
| max_iter | 5000 |
| n_jobs | $-1$ |
| random_state | 42 |

The reported accuracies in Table 1 are mean and standard deviation across the 10 outer folds.

# E. Compute Environment

All experiments were run on the TU Delft Delft AI Cluster (Delft AI Cluster (DAIC), 2024) (`general` partition, `long` QoS), submitted via SLURM. Each job was allocated 20 CPU cores on a single node, with 5 GB of RAM per core (100 GB total) and a maximum wall time of 120 hours. Nodes in the `general` partition use Intel Xeon-class CPUs; the specific SKU varies by node assignment. Tree construction used the 20 allocated cores via RapidNJ's parallel implementation, with UPGMA completing in approximately 11 hours and Neighbor-Joining in approximately 3 days on the full 103,883-sample dataset.

# F. Outlier Detection Methodology

This section details our approach to identifying and removing outliers from phylogenetic trees before temporal validation.

### F.1. Motivation

Outliers significantly distort phylogenetic tree topology by artificially pushing Most Recent Common Ancestors (MRCAs) toward the root. This inflation of distances corrupts temporal comparisons and evolutionary inferences.

Such outliers often arise from two sources: (1) **mislabeled samples** where VirusTotal vendor consensus incorrectly assigns a sample to a family, or (2) **highly divergent variants** that underwent extreme modification. In both cases, these samples do not represent typical family evolution and distort inter-family relationships. Our two-stage outlier detection and removal process ensures clean tree structures for reliable validation.

We identify outliers based on *statistical distance within their labeled family*, independent of temporal consistency. We are not removing samples because they violate our temporal validation; rather, we are removing samples that are statistical anomalies within their own family, then validating the resulting tree structure.

### F.2. Lateral Distance Calculation

For each family $\mathcal{F}$, we compute *lateral distances* between all pairs of leaves within that family. The lateral distance represents the evolutionary distance between two samples, measured as the sum of branch lengths through their shared ancestor.

F.2.1. DEFINITION

The lateral distance between two leaves $s_i$ and $s_j$ is:

$$d_{\text{lateral}}(s_i, s_j) = d_{\mathcal{T}}(s_i, a) + d_{\mathcal{T}}(a, s_j) \tag{5}$$

where $a = \text{MRCA}(s_i, s_j)$ is the Most Recent Common Ancestor of $s_i$ and $s_j$, and $d_{\mathcal{T}}(\cdot, \cdot)$ denotes the sum of branch lengths along the path in tree $\mathcal{T}$.

F.2.2. VISUAL EXAMPLE

Figure 8 illustrates this calculation for a simple case.

**Family WpBruteBot**

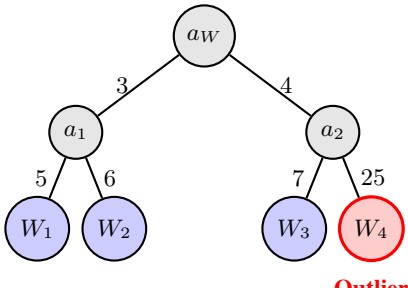

*Figure 8.* Lateral distance calculation for family WpBruteBot. The distance between leaves $W_1$ and $W_2$ is computed as the sum of branch lengths through their shared ancestor $a_W$.

## F.3. Median Lateral Distance

For each sample $s_i \in \mathcal{F}$, we compute its distance to all other samples in the family and extract the median value. This median serves as a robust summary of how far the sample is from other members of its family.

F.3.1. DEFINITION

For sample $s_i$ in family $\mathcal{F}$:

$$\tilde{d}_i = \text{median}\left\{ d_{\text{lateral}}(s_i, s_j) : s_j \in \mathcal{F}, s_j \neq s_i \right\} \tag{6}$$

F.3.2. STEP-BY-STEP EXAMPLE

Consider a family with four samples as illustrated in Figure 9.

**Family WpBruteBot**

| Sample | Distances | Median $\tilde{d}_i$ | Status |
|--------|-----------|----------------------|--------|
| $W_1$ | $[11, 19, 37]$ | 19 | Normal |
| $W_2$ | $[11, 20, 38]$ | 20 | Normal |
| $W_3$ | $[19, 20, 32]$ | 20 | Normal |
| $W_4$ | $[37, 38, 32]$ | **37** | **Outlier** |

*Figure 9.* Median lateral distance calculation. Sample $W_4$ has substantially higher median lateral distance compared to others, indicating it is an outlier.

## F.4. IQR-Based Outlier Threshold

We apply a standard statistical method for outlier detection based on the Interquartile Range (IQR), commonly used in boxplot analysis (Tukey, 1977).

### F.4.1. QUARTILE CALCULATIONS

Given the set of median lateral distances $\{\tilde{d}_1, \tilde{d}_2, \ldots, \tilde{d}_{|\mathcal{F}|}\}$ for family $\mathcal{F}$:

- **First Quartile ($Q_1$):** The 25th percentile of median lateral distances

- **Third Quartile ($Q_3$):** The 75th percentile of median lateral distances

- **Interquartile Range (IQR):** $\mathrm{IQR} = Q_3 - Q_1$

### F.4.2. OUTLIER THRESHOLD

A sample $s_i$ is flagged as an outlier if:

$$\tilde{d}_i > Q_3 + 1.5 \times \mathrm{IQR} \tag{7}$$

This threshold, widely used in statistical analysis, identifies values that lie significantly beyond the typical range of the data.

## F.5. Impact of Outliers on Tree Topology

Outliers fundamentally alter phylogenetic tree structure by distorting MRCA placement. Figure 10 illustrates how an outlier forces the MRCA deeper in the tree.

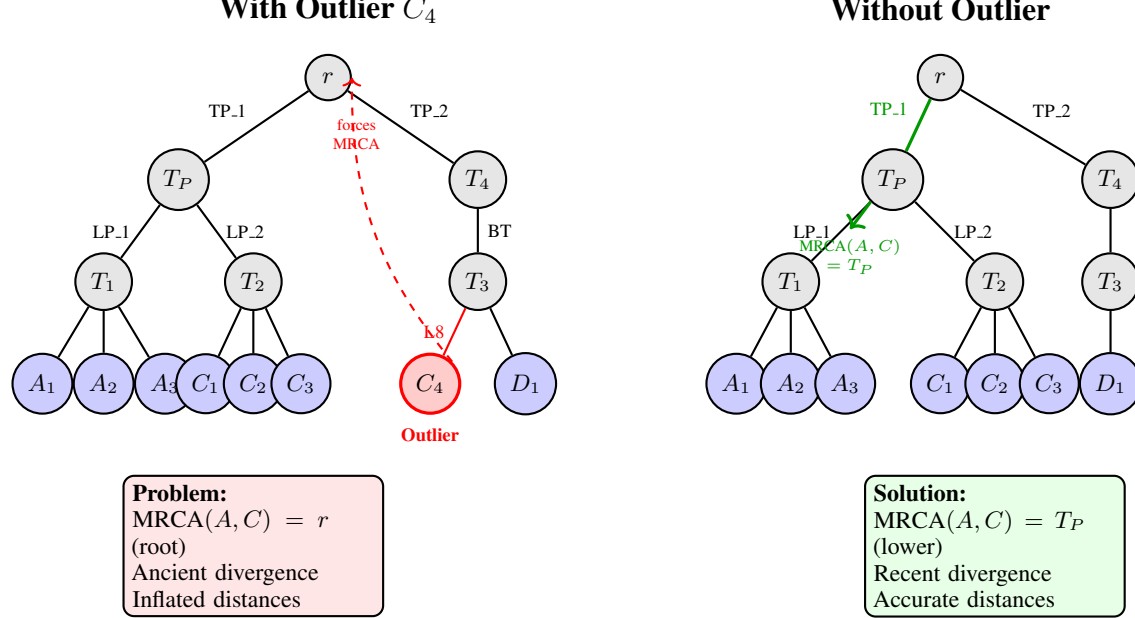

*Figure 10.* Impact of outliers on tree topology. **Left:** With outlier $C_4$ present, its extreme divergence from other Family C members ($C_1, C_2, C_3$) causes Family C to be placed far from Family A during tree construction. Families A and C share MRCA at root $r$, inflating distances. **Right:** After removing $C_4$ and rebuilding, the remaining Family C samples ($C_1, C_2, C_3$) cluster with Family A, sharing MRCA at intermediate node $T_P$, yielding accurate distances.

## F.6. Two-Stage Tree Construction Algorithm

Our outlier handling employs a two-stage process that ensures clean tree structure for subsequent temporal validation.

F.6.1. STAGE 1: DETECTION

---

**Algorithm 1** Outlier Detection from Initial Tree

---

1: **Input:** Dataset $\mathcal{S}$ with family labels
2: **Output:** Set of outlier samples $\mathcal{O}$
3:
4: Construct initial tree $\mathcal{T}_{\text{init}} \leftarrow \text{BuildTree}(\mathcal{S})$
5: Initialize $\mathcal{O} \leftarrow \emptyset$
6:
7: **for all** family $\mathcal{F}$ in dataset **do**
8:      $L_{\mathcal{F}} \leftarrow \{\text{leaves in } \mathcal{T}_{\text{init}} \text{ from } \mathcal{F}\}$
9:      **if** $|L_{\mathcal{F}}| < 2$ **then**
10:          **continue**
11:      **end if**
12:
13:      **for all** $s_i \in L_{\mathcal{F}}$ **do**
14:          $D_i \leftarrow []$
15:          **for all** $s_j \in L_{\mathcal{F}}, s_j \neq s_i$ **do**
16:              $a \leftarrow \text{MRCA}(s_i, s_j) \text{ in } \mathcal{T}_{\text{init}}$
17:              $d \leftarrow d_{\mathcal{T}}(s_i, a) + d_{\mathcal{T}}(a, s_j)$
18:              Append $d$ to $D_i$
19:          **end for**
20:          $\tilde{d}_i \leftarrow \text{median}(D_i)$
21:      **end for**
22:
23:      $Q_1 \leftarrow \text{percentile}_{25}(\{\tilde{d}_i\})$
24:      $Q_3 \leftarrow \text{percentile}_{75}(\{\tilde{d}_i\})$
25:      threshold $\leftarrow Q_3 + 1.5 \times (Q_3 - Q_1)$
26:
27:      **for all** $s_i \in L_{\mathcal{F}}$ **do**
28:          **if** $\tilde{d}_i > \text{threshold}$ **then**
29:              $\mathcal{O} \leftarrow \mathcal{O} \cup \{s_i\}$
30:          **end if**
31:      **end for**
32: **end for**
33:
34: **return** $\mathcal{O}$

---

F.6.2. STAGE 2: RECONSTRUCTION

---

**Algorithm 2** Clean Tree Reconstruction

---

1: **Input:** Dataset $\mathcal{S}$, outlier set $\mathcal{O}$
2: **Output:** Clean tree $\mathcal{T}_{\text{clean}}$
3:
4: $\mathcal{S}_{\text{clean}} \leftarrow \mathcal{S} \setminus \mathcal{O}$
5: $\mathcal{T}_{\text{clean}} \leftarrow \text{BuildTree}(\mathcal{S}_{\text{clean}})$
6:
7: **return** $\mathcal{T}_{\text{clean}}$

---

# G. Temporal Validation Framework

This section provides detailed explanation of our temporal validation methodology, including the path-length assumption and why it works without requiring a strict molecular clock.

## G.1. The Path-Length Assumption

Our temporal validation rests on a fundamental assumption about the relationship between branch length and divergence order (Felsenstein, 2004).

### G.1.1. FORMAL STATEMENT

For two samples $s_i$ and $s_j$ sharing a Most Recent Common Ancestor (MRCA) $a$:

$$L_i < L_j \implies s_i \text{ emerged earlier than } s_j \tag{8}$$

where $L_i = d_{\mathcal{T}}(s_i, a)$ denotes the path length from leaf $s_i$ to ancestor $a$.

### G.1.2. INTUITION

Consider the evolutionary process illustrated in Figure 11:

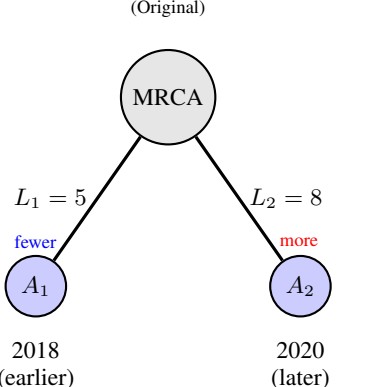

*Figure 11.* Path-length intuition. Sample $A_1$ with shorter branch has fewer modifications from ancestor, suggesting earlier emergence. Sample $A_2$ with longer branch has more modifications, suggesting later emergence.

## G.2. Global Clock vs. Local Clock

A critical distinction: our path-length assumption does *not* require a global molecular clock (Bromham & Penny, 2003). It assumes only local rate homogeneity between the immediate siblings we compare.

### G.2.1. WHAT A GLOBAL MOLECULAR CLOCK ASSUMES

A global molecular clock assumes a constant rate across *all* lineages (Zuckerkandl & Pauling, 1965):

$$\text{branch length} = \text{rate} \times \text{time} \tag{9}$$

UPGMA assumes this and performs poorly when it is violated (Sokal & Michener, 1958). Our embedding drift analysis (Figure 5 in the main text) shows malware families violate it: some families drift over ten times faster than others.

### G.2.2. WHAT OUR ASSUMPTION REQUIRES

We never invoke rate constancy across the tree. We invoke it only locally, between two immediate siblings sharing a parent. For such a pair $(s_i, s_j)$:

$$L_i < L_j \implies t(s_i) < t(s_j) \tag{10}$$

This holds when the two siblings accumulate structural change at comparable rates, so that the longer branch reflects more elapsed development rather than a faster mutation tempo. Crucially, this is a statement about one sibling pair, not about the whole tree: each pair carries its own local rate, and rates may differ freely across pairs and across families. In phylogenetic terms this is a local clock rather than a global one, a substantially weaker and more defensible assumption.

We do not rely on developer intent or monotonic complexity growth. Instead, we interpret branch length as relative structural divergence in embedding space, and we validate empirically that, within shallow divergences, shorter branches tend to carry earlier VirusTotal timestamps.

### G.2.3. WHY A LOCAL CLOCK IS PLAUSIBLE FOR SIBLINGS

The local clock is credible precisely because we restrict comparisons to immediate siblings. Such variants typically share an author or development group, a common toolchain, and a similar modification cadence, so comparable rates between them is a mild assumption rather than a sweeping one. The failure mode is explicit: if two siblings were developed at very different tempos, the faster-drifting variant can acquire the longer branch despite emerging earlier, inverting the inferred order. Restricting to shallow divergences limits the structural and temporal gap over which such rate differences can accumulate, which is why these comparisons are the most reliable (Section G.3).

### G.2.4. WHY THIS WORKS WITH VARIABLE RATES ACROSS FAMILIES

A global clock fails on Figure 5's ten-fold rate variation, but our local-clock ordering does not, because we never compare across families with different rates. We compare siblings within a family, and the ordering depends only on the relative branch lengths inside that pair:

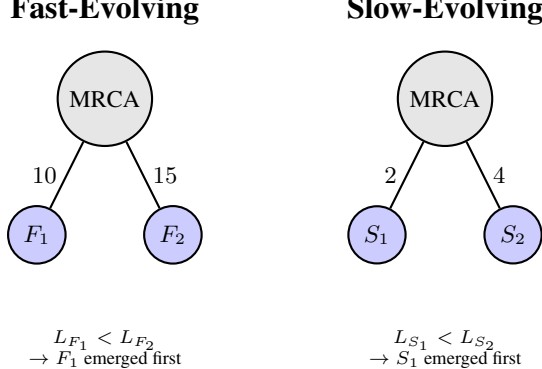

*Figure 12.* Variable mutation rates across families. The fast-evolving and slow-evolving families operate at different absolute rates, yet within each family the relative branch lengths still recover divergence order. Our ordering requires only this within-pair (local) consistency, not equal rates across families.

**Key insight:** We compare siblings *within* the same family, where both variants were likely produced by the same author(s) or group under similar development practices. This local consistency is what the path-length ordering needs, and it is far weaker than the global rate constancy a molecular clock demands.

### G.3. Why Siblings Are Most Reliable

Our validation focuses on *shallow divergences* (immediate siblings) rather than deep comparisons, following recommendations from Felsenstein (1985).

### G.3.1. MEASUREMENT RELIABILITY

Short paths are more reliable because they:

- Accumulate less estimation error in branch lengths

- Exhibit less rate variation over time

- Are less susceptible to convergent evolution

- Avoid saturation from multiple mutations at the same site (Felsenstein, 2004)

### G.3.2. TEMPORAL RESOLUTION

Siblings represent recent divergences where:

- Timestamps are directly comparable (similar time scales)

- Evolutionary changes are minimal (clearer signal)

- Emergence order is unambiguous

- Development context is shared (same authors, tools, methods)

### G.4. Validation Procedure

For each leaf:

1. Identify immediate parent node $a$

2. Find all sibling pairs $(s_i, s_j)$ descending from $a$

3. For each pair:
   - Compute $L_i = d_{\mathcal{T}}(s_i, a)$ and $L_j = d_{\mathcal{T}}(s_j, a)$
   - Get timestamps $t(s_i)$ and $t(s_j)$ from VirusTotal
   - Check: $(L_i < L_j \wedge t(s_i) < t(s_j))$ or $(L_i > L_j \wedge t(s_i) > t(s_j))$

4. Temporal consistency = proportion of consistent pairs

Our 87% temporal consistency confirms the assumption holds in practice.

## H. Inter-Family Analysis

This section explains our methodology for inferring evolutionary relationships between malware families and constructing the inter-family graphs shown in case studies.

### H.1. Motivation

When comparing different families, their shared MRCA is typically a deep internal node. Individual sample-to-sample comparisons are unreliable due to long paths and within-family variation. We therefore compare families as aggregates using robust statistics, producing a directed graph where edges represent inferred evolutionary relationships.

### H.2. Methodology

The inter-family analysis proceeds in three stages: distance calculation, graph construction, and graph simplification.

### H.2.1. STAGE 1: DISTANCE CALCULATION

For each pair of families $\mathcal{F}_A$ and $\mathcal{F}_B$:

1. Identify their shared MRCA: $r = \text{MRCA}(\mathcal{F}_A, \mathcal{F}_B)$

2. Compute the path distance from $r$ to each leaf in both families

3. Calculate the median distance for each family:

$$\tilde{d}_A = \text{median}\{d_{\mathcal{T}}(s,r) : s \in \mathcal{F}_A\} \tag{11}$$

$$\tilde{d}_B = \text{median}\{d_{\mathcal{T}}(s,r) : s \in \mathcal{F}_B\} \tag{12}$$

The median is chosen for robustness (Rousseeuw & Croux, 1993): each family contains many samples at varying distances, and the mean would be sensitive to extreme values. The median represents the "typical" evolutionary depth for each family.

### H.2.2. STAGE 2: GRAPH CONSTRUCTION

We construct a directed graph where nodes represent families and edges represent evolutionary relationships. For each family pair $(\mathcal{F}_A, \mathcal{F}_B)$:

- If $\tilde{d}_A < \tilde{d}_B$: create edge $\mathcal{F}_A \rightarrow \mathcal{F}_B$ with weight $w = \tilde{d}_A$
- If $\tilde{d}_B < \tilde{d}_A$: create edge $\mathcal{F}_B \rightarrow \mathcal{F}_A$ with weight $w = \tilde{d}_B$

The edge direction encodes evolutionary precedence: edges point *from* the earlier-diverging family (shorter median distance to MRCA) *to* the later-diverging family. The edge weight equals the source family's median distance to the shared MRCA.

### H.2.3. STAGE 3: GRAPH SIMPLIFICATION

The raw graph contains $O(K^2)$ edges for $K$ families, obscuring primary lineages. We simplify by retaining only the minimum-weight outgoing edge from each node:

$$\text{For each node } n : \text{ keep only } \arg\min_{e \in \text{outgoing}(n)} w(e) \tag{13}$$

This focuses the visualization on the strongest (lowest-weight) evolutionary connections while preserving each family's most likely progenitor relationship.

### H.3. Interpreting Edge Weights

**Edge weights represent the source family's median Euclidean distance (in embedding space) to the shared MRCA with the target family.** This has several implications:

- **Lower weights indicate closer relationships**: A weight of $w = 9.6$ (e.g., Mirai $\rightarrow$ Bashlite) indicates the source family's samples are, on average, closer to their shared ancestor than a weight of $w = 30.0$ (e.g., Mirai $\rightarrow$ Conti).

- **Weights are not symmetric**: The edge $A \rightarrow B$ has weight $\tilde{d}_A$, while a hypothetical edge $B \rightarrow A$ would have weight $\tilde{d}_B$. Only the edge from the closer family is created.

- **High weights suggest low confidence**: In our experiments, validated relationships typically exhibit weights below 25, while connections lacking threat intelligence corroboration often exceed this threshold. However, weight magnitude alone does not determine relationship validity (see SmokeLoader case study in Appendix I).

### H.4. Visual Example

Figure 13 illustrates the edge weight calculation for two families.

### H.5. Distinction from Minimum Spanning Tree Approaches

Prior work by Cozzi et al. (2020) employed Minimum Spanning Trees (MST) as proxies for phylogenetic analysis of IoT malware. Our graph simplification procedure differs fundamentally from MST construction:

- **MST approach**: Selects edges that minimize *total* edge weight globally, producing an undirected tree connecting all nodes with minimum sum of weights. This optimizes for global parsimony but does not encode evolutionary directionality.

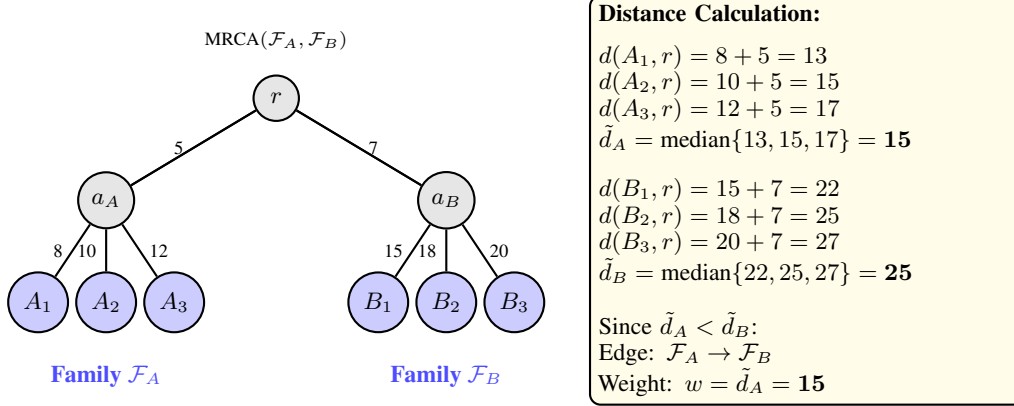

**Distance Calculation:**

$d(A_1, r) = 8 + 5 = 13$
$d(A_2, r) = 10 + 5 = 15$
$d(A_3, r) = 12 + 5 = 17$
$\tilde{d}_A = \text{median}\{13, 15, 17\} = \mathbf{15}$

$d(B_1, r) = 15 + 7 = 22$
$d(B_2, r) = 18 + 7 = 25$
$d(B_3, r) = 20 + 7 = 27$
$\tilde{d}_B = \text{median}\{22, 25, 27\} = \mathbf{25}$

Since $\tilde{d}_A < \tilde{d}_B$:
Edge: $\mathcal{F}_A \rightarrow \mathcal{F}_B$
Weight: $w = \tilde{d}_A = \mathbf{15}$

*Figure 13.* Edge weight calculation example. Family $\mathcal{F}_A$ has median distance 15 to the shared MRCA; Family $\mathcal{F}_B$ has median distance 25. The resulting edge points from $\mathcal{F}_A$ to $\mathcal{F}_B$ with weight 15.

- **Our approach**: Retains the *minimum-weight outgoing edge from each node*, preserving edge directionality (from earlier-diverging to later-diverging families). Each family points to its single most likely progenitor based on phylogenetic distance, rather than optimizing a global objective.

These approaches yield structurally different graphs. MST produces an undirected tree where high-degree hub nodes emerge from global optimization, while our method produces a directed graph where each family has exactly one outgoing edge to its nearest evolutionary neighbor. Our approach thus prioritizes interpretability of individual lineage relationships over global topological optimality, and explicitly encodes the temporal directionality central to evolutionary analysis.

### H.6. Algorithms

The complete algorithms for inter-family analysis are provided in Appendix K:

- **Algorithm 7**: Computes median distances from each family's leaves to their shared MRCA

- **Algorithm 8**: Constructs directed edges from earlier-diverging to later-diverging families, with edge weight equal to the source family's median distance

- **Algorithm 9**: Retains only minimum-weight outgoing edge per node to focus on primary lineages

- **Algorithm 11**: Optionally removes intra-family outliers before distance calculation (used for cleaner inter-family analysis)

## I. Case Studies

### I.1. Case Study: SmokeLoader

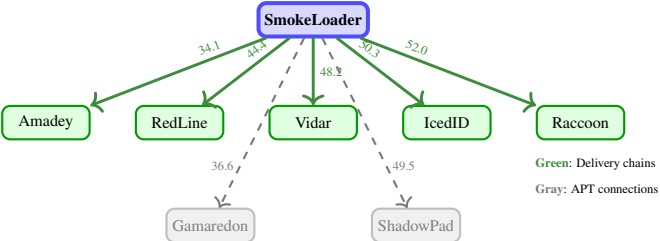

*Figure 14.* SmokeLoader inter-family subgraph with edge weights. Green edges denote validated delivery chain associations; gray dashed edges indicate connections to APT families.

SmokeLoader is a modular backdoor active since 2011 that remains prevalent in the cybercrime ecosystem (Huntress Labs, 2024). Unlike Mirai's code evolution relationships, SmokeLoader's phylogenetic connections represent *delivery chain associations* arising from its role as a loader distributing secondary payloads, not code inheritance. Figure 14 shows the inter-family subgraph with edge weights. MalTree identifies five associations (green) that align with documented delivery relationships: **Amadey** ($w = 34.1$), distributed through SmokeLoader via malicious crack websites since 2022 (AhnLab ASEC, 2024; Toulas, 2022); **RedLine** ($w = 44.4$) and **Vidar** ($w = 48.2$), documented by CYFIRMA alongside the STOP/Djvu ransomware ecosystem (CYFIRMA Research, 2024); **IcedID** ($w = 50.3$), corroborated by the 2024 Operation Endgame takedown (Proofpoint Threat Research, 2024); and **Raccoon** ($w = 52.0$), confirmed by VMRay as a SmokeLoader payload (VMRay, 2024). These families were *delivered by* SmokeLoader, not *derived from* it; phylogenetic proximity reflects shared packing techniques, evasion mechanisms, and infrastructure patterns rather than code ancestry.

The graph also reveals connections to Gamaredon ($w = 36.6$) and ShadowPad ($w = 49.5$), both nation-state APT tools with separate development lineages. These connections arise because SmokeLoader's generic loader functionality (network communications, file operations, persistence mechanisms) creates feature-space proximity with families implementing similar primitives. This illustrates a limitation of the current approach: it cannot automatically distinguish code evolution from delivery chain associations or incidental feature-space proximity. Loader families may exhibit connections reflecting operational co-occurrence rather than evolutionary ancestry, and practitioners should cross-reference phylogenetic results with threat intelligence when analyzing hub families. For defenders, delivery chain associations remain operationally valuable, as SmokeLoader activity suggests elevated probability of subsequent stealer deployment.

### I.2. Case Study: Hub Family

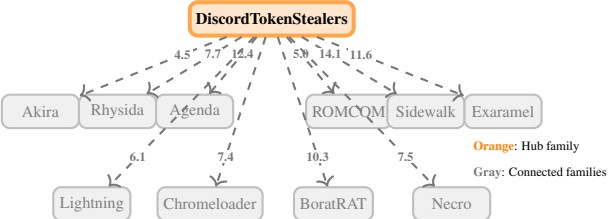

*Figure 15.* DiscordTokenStealers hub pattern showing 10 representative connections (of 22 total); edge weights in the figure range from $w = 4.5$ to $w = 14.1$.

The DiscordTokenStealers family exhibits a hub pattern with 22 inferred connections. This family comprises credential-harvesting tools targeting Discord authentication tokens, implementing file system traversal and HTTP exfiltration. Figure 15 shows a subset of these connections. MalTree links DiscordTokenStealers to ransomware families including Akira ($w = 4.5$), Rhysida ($w = 7.7$), and Agenda ($w = 12.4$); to nation-state APT tools including ROMCOM ($w = 5.0$), Sidewalk ($w = 14.1$), and Exaramel ($w = 11.6$); and to other malware including Chromeloader ($w = 7.4$), BoratRAT ($w = 10.3$), and Necro ($w = 7.5$). These connections arise because DiscordTokenStealers' generic functionality (credential harvesting, file operations, HTTP communications) overlaps with common malware primitives shared across diverse families, creating proximity in the embedding space. The low edge weights are comparable to validated Mirai variants ($w = 9.6$–$20.8$), indicating that weight magnitude alone does not distinguish meaningful relationships from incidental feature overlap. This illustrates a limitation: families implementing widely-shared primitives may exhibit hub patterns that reflect functional similarity rather than evolutionary or operational relationships, and practitioners should apply additional scrutiny when interpreting connections from such families.

## I.3. Case Study: Scattered Ecosystems

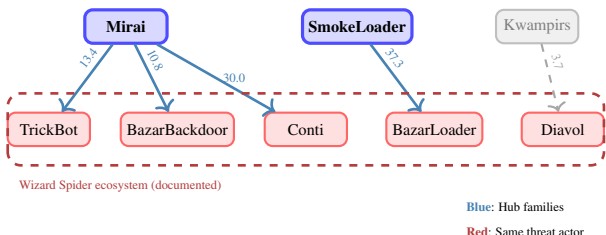

*Figure 16.* Conti/TrickBot ecosystem fragmentation. Red nodes represent the documented Wizard Spider threat actor's toolset, which should cluster together but instead scatter across unrelated hub families. Dashed box indicates the expected phylogenetic grouping.

The Conti ransomware ecosystem represents a well-documented threat actor cluster that MalTree fails to recover. The Wizard Spider group operated TrickBot (banking trojan turned loader), BazarBackdoor and BazarLoader (TrickBot successors), Conti (ransomware-as-a-service), and Diavol (secondary ransomware strain), with extensive code sharing and operational overlap documented by CISA, Mandiant, and the 2022 Conti Leaks (Cybersecurity and Infrastructure Security Agency, 2021). Figure 16 shows how these families appear in the MalTree graph: rather than clustering together, they scatter across unrelated hub families. TrickBot ($w = 13.4$), BazarBackdoor ($w = 10.8$), and Conti ($w = 30.0$) all connect through Mirai, an IoT botnet with no operational relationship to Windows-targeting crimeware. BazarLoader connects through SmokeLoader ($w = 37.3$), plausibly reflecting delivery chain co-occurrence but not the direct code lineage with BazarBackdoor. Most notably, Diavol connects through Kwampirs ($w = 3.7$), an unrelated healthcare-sector RAT.

This fragmentation occurs because the Conti ecosystem, despite shared authorship, exhibits substantial functional diversity: TrickBot's banking trojan origins, Bazar's loader functionality, and Conti/Diavol's ransomware payloads occupy different regions of the embedding space. The phylogenetic algorithm, optimizing for feature similarity rather than threat actor attribution, routes each family through the nearest hub. This case study illustrates a fundamental limitation: MalTree captures *functional* rather than *organizational* relationships. Threat actor attribution requires additional intelligence sources beyond code similarity, and practitioners should not expect phylogenetic methods to recover actor-level clustering when toolsets span multiple functional categories.

## J. Embedding Drift Analysis

This appendix details the methodology for computing embedding drift rates used to assess whether UPGMA's molecular clock assumption holds for malware evolution.

### J.1. Motivation

UPGMA assumes a molecular clock: all lineages evolve at a constant rate. If this assumption holds, embedding distances should increase proportionally with time elapsed between sample emergence. We measure drift rate (distance per year) across families; high variance indicates non-uniform evolution, favoring Neighbor-Joining over UPGMA.

### J.2. Drift Rate Definition

Let $s_i \in \mathcal{F}$ be a sample with embedding $\mathbf{e}_i \in \mathbb{R}^d$ and first-submission year $y_i$. For two samples $s_i$ and $s_j$ where $y_j > y_i$, the *drift rate* is:

$$r_{ij} = \frac{\|\mathbf{e}_i - \mathbf{e}_j\|_2}{y_j - y_i} \tag{14}$$

This normalizes Euclidean distance by time difference in years, yielding units of embedding distance per year.

## J.3. Aggregation Per Family

For each family $\mathcal{F}$ spanning multiple years, we compute drift rates for all valid sample pairs:

$$R_{\mathcal{F}} = \{r_{ij} : s_i, s_j \in \mathcal{F}, \ y_j > y_i\} \tag{15}$$

We extract the minimum and maximum drift rates:

$$r_{\mathcal{F}}^{\min} = \min(R_{\mathcal{F}}) \tag{16}$$
$$r_{\mathcal{F}}^{\max} = \max(R_{\mathcal{F}}) \tag{17}$$

These represent the range of evolutionary rates observed within each family. The complete procedure is formalized in Appendix K.

# K. Validation Algorithms

## K.1. Virustotal Timestamps

---

**Algorithm 3** Validate Chronological Order of Phylogenetic Tree Leaves

---

1: **Input:** Phylogenetic tree $\mathcal{T}$, timestamps for each leaf
2: **Output:** Proportion of correctly ordered leaf pairs
3:
4: $correctPairs \leftarrow 0$
5: $totalComparisons \leftarrow 0$
6:
7: **for all** leaf $\ell$ in $\mathcal{T}$ **do**
8:     $a \leftarrow \text{GETDIRECTANCESTOR}(\ell)$
9:     $L_a \leftarrow \text{GETLEAVESFROMANCESTOR}(a)$
10:
11:     **for all** $(\ell_1, \ell_2) \in L_a$ **do**
12:         $d_1 \leftarrow d_{\mathcal{T}}(a, \ell_1)$
13:         $d_2 \leftarrow d_{\mathcal{T}}(a, \ell_2)$
14:         $t_1 \leftarrow timestamp[\ell_1]$
15:         $t_2 \leftarrow timestamp[\ell_2]$
16:
17:         **if** $(d_1 < d_2 \wedge t_1 < t_2) \vee (d_1 > d_2 \wedge t_1 > t_2)$ **then**
18:             $correctPairs \leftarrow correctPairs + 1$
19:         **end if**
20:         $totalComparisons \leftarrow totalComparisons + 1$
21:     **end for**
22: **end for**
23:
24: $accuracy \leftarrow correctPairs/totalComparisons$
25: **return** $accuracy$

---

## K.2. Embedding Drift Analysis

---

**Algorithm 4** Calculate Drift Rates

---

1: **Input:** Distance matrix $\mathcal{D}$, family-variant binning by year
2: **Output:** Drift rate statistics (distance per year) per family and globally
3:
4: $familyRates \leftarrow \{\}$
5: $r_{\min}^{global} \leftarrow \infty, r_{\max}^{global} \leftarrow -\infty$
6:
7: **for all** family $\mathcal{F}$ with year data $Y_{\mathcal{F}}$ **do**
8:     Sort years in $Y_{\mathcal{F}}$
9:     $r_{\min}^{\mathcal{F}} \leftarrow \infty, r_{\max}^{\mathcal{F}} \leftarrow -\infty$
10:
11:     **for all** year $y_1 \in Y_{\mathcal{F}}$ except last **do**
12:         **for all** year $y_2 \in Y_{\mathcal{F}}$ where $y_2 > y_1$ **do**
13:             **for all** variant $v_1 \in Y_{\mathcal{F}}[y_1]$ **do**
14:                 $\tau_1 \leftarrow \text{FORMATTAXON}(\mathcal{F}, v_1)$
15:                 **for all** variant $v_2 \in Y_{\mathcal{F}}[y_2]$ **do**
16:                     $\tau_2 \leftarrow \text{FORMATTAXON}(\mathcal{F}, v_2)$
17:                     **if** $(\tau_1, \tau_2) \in \mathcal{D}$ **then**
18:                         $d \leftarrow \mathcal{D}[\tau_1, \tau_2]$
19:                         $r \leftarrow d/(y_2 - y_1)$
20:                         Update $r_{\min}^{\mathcal{F}}, r_{\max}^{\mathcal{F}}, r_{\min}^{global}, r_{\max}^{global}$ using $r$
21:                     **end if**
22:                 **end for**
23:             **end for**
24:         **end for**
25:     **end for**
26:     Store $(r_{\min}^{\mathcal{F}}, r_{\max}^{\mathcal{F}})$ in $familyRates[\mathcal{F}]$
27: **end for**
28:
29: **return** $familyRates, r_{\min}^{global}, r_{\max}^{global}$

---

## K.3. Inter-family Analysis – Without Outlier Thresholding

---

**Algorithm 5** Analyze Phylogenetic Tree

---

1: **Input:** Tree file path
2: **Output:** Simplified graph $G$ representing evolutionary relationships
3:
4: $\mathcal{T}, \mathcal{M} \leftarrow \text{LOADTREEANDAGGREGATELEAVES}(path)$
5: $\mathcal{D} \leftarrow \text{CALCULATEDISTANCES}(\mathcal{T}, \mathcal{M})$
6: $G \leftarrow \text{BUILDGRAPH}(\mathcal{D})$
7: $G_{simple} \leftarrow \text{SIMPLIFYGRAPH}(G)$
8: **return** $G_{simple}$

---

---

**Algorithm 6** Load Tree and Aggregate Leaves

---

1: **Input:** Tree file path
2: **Output:** Tree $\mathcal{T}$, mapping $\mathcal{M} : family \rightarrow leaves$
3:
4: $\mathcal{T} \leftarrow \text{LOADTREE}(path)$
5: $\mathcal{M} \leftarrow \{\}$
6: **for all** leaf $\ell \in \mathcal{T}$ **do**
7:     $\mathcal{F} \leftarrow \text{EXTRACTFAMILYNAME}(\ell)$
8:     **if** $\mathcal{F} \notin \mathcal{M}$ **then**
9:         $\mathcal{M}[\mathcal{F}] \leftarrow []$
10:    **end if**
11:    Append $\ell$ to $\mathcal{M}[\mathcal{F}]$
12: **end for**
13: **return** $\mathcal{T}, \mathcal{M}$

---

**Algorithm 7** Calculate Inter-family Distances

---

1: **Input:** Tree $\mathcal{T}$, family-to-leaves mapping $\mathcal{M}$
2: **Output:** Distance matrix $\mathcal{D}$
3:
4: $\mathcal{D} \leftarrow \{\}$
5: $families \leftarrow \text{KEYS}(\mathcal{M})$
6: **for all** $\mathcal{F}_1 \in families$ **do**
7:     **for all** $\mathcal{F}_2 \in families$ where $\mathcal{F}_2 \neq \mathcal{F}_1$ **do**
8:         $a \leftarrow \text{MRCA}(\mathcal{F}_1, \mathcal{F}_2)$
9:         $D_1 \leftarrow [d_{\mathcal{T}}(a, \ell) \text{ for } \ell \in \mathcal{M}[\mathcal{F}_1]]$
10:       $D_2 \leftarrow [d_{\mathcal{T}}(a, \ell) \text{ for } \ell \in \mathcal{M}[\mathcal{F}_2]]$
11:       $\mathcal{D}[(\mathcal{F}_1, \mathcal{F}_2)] \leftarrow (\text{MEDIAN}(D_1), \text{MEDIAN}(D_2))$
12:    **end for**
13: **end for**
14: **return** $\mathcal{D}$

---

**Algorithm 8** Build Directed Graph from Distances

---

1: **Input:** Distance matrix $\mathcal{D}$
2: **Output:** Directed graph $G$
3:
4: $G \leftarrow \text{NEWGRAPH}()$
5: **for all** $(\mathcal{F}_1, \mathcal{F}_2, (d_1, d_2)) \in \mathcal{D}$ **do**
6:     **if** $d_1 < d_2$ **then**
7:         $\text{ADDEDGE}(G, \mathcal{F}_1, \mathcal{F}_2, d_1)$
8:     **else**
9:         $\text{ADDEDGE}(G, \mathcal{F}_2, \mathcal{F}_1, d_2)$
10:    **end if**
11: **end for**
12: **return** $G$

---

---

**Algorithm 9** Simplify Graph by Retaining Minimum-Weight Outgoing Edge per Node

---

1: **Input:** Graph $G$
2: **Output:** Simplified graph $G'$
3:
4: $G' \leftarrow$ NEWGRAPH()
5: **for all** node $n \in G$ **do**
6:     $e_{\min} \leftarrow$ FINDMINEDGE($n$)
7:     ADDEDGE($G', n, e_{\min}.target, e_{\min}.weight$)
8: **end for**
9: **return** $G'$

---

## K.4. Inter-family Analysis – With Outlier Thresholding

---

**Algorithm 10** Analyze Phylogenetic Tree with Outlier Removal

---

1: **Input:** Tree file path
2: **Output:** Simplified graph $G$ with outliers removed
3:
4: $\mathcal{T}, \mathcal{M} \leftarrow$ LOADTREEANDAGGREGATELEAVES(*path*)
5: $\mathcal{M}' \leftarrow$ REMOVEINTRAFAMILYOUTLIERS($\mathcal{T}, \mathcal{M}$)
6: $\mathcal{D} \leftarrow$ CALCULATEDISTANCES($\mathcal{T}, \mathcal{M}'$)
7: $G \leftarrow$ BUILDGRAPH($\mathcal{D}$)
8: $G_{simple} \leftarrow$ SIMPLIFYGRAPH($G$)
9: **return** $G_{simple}$

---

**Algorithm 11** Remove Intra-family Outliers via IQR

---

1: **Input:** Tree $\mathcal{T}$, family-to-leaves mapping $\mathcal{M}$
2: **Output:** Filtered mapping $\mathcal{M}'$
3:
4: $\mathcal{M}' \leftarrow \{\}$
5: **for all** $(\mathcal{F}, L) \in \mathcal{M}$ **do**
6:     **if** $|L| \geq 2$ **then**
7:         $D \leftarrow$ COMPUTELEAFTOLEAFDISTANCES($\mathcal{T}, L$)
8:         $\tilde{D} \leftarrow$ MEDIANPERLEAF($D$)
9:         $IQR \leftarrow$ COMPUTEIQR($\tilde{D}$)
10:        $\mathcal{M}'[\mathcal{F}] \leftarrow$ FILTERBYIQR($\tilde{D}, IQR$)
11:     **else**
12:        $\mathcal{M}'[\mathcal{F}] \leftarrow L$
13:     **end if**
14: **end for**
15: **return** $\mathcal{M}'$

---

# L. Embedding Architecture Details

This appendix provides architecture and hyperparameter details for the embedding extraction and fusion pipeline.

## L.1. Image Embedding Network

We employ a two-stage transfer learning approach for image embeddings (Figure 17). Model V1 is a ResNet-50 initialized with ImageNet weights and trained on public malware image datasets (MalImg, MaleVis, MalNet), achieving 85% accuracy. Model V2 inherits V1's weights and is fine-tuned on our dataset with a modified classification layer (538 classes), achieving 95% accuracy. Images are resized to $224 \times 224$ pixels and normalized using ImageNet statistics: mean $[0.485, 0.456, 0.406]$ and standard deviation $[0.229, 0.224, 0.225]$ across RGB channels.

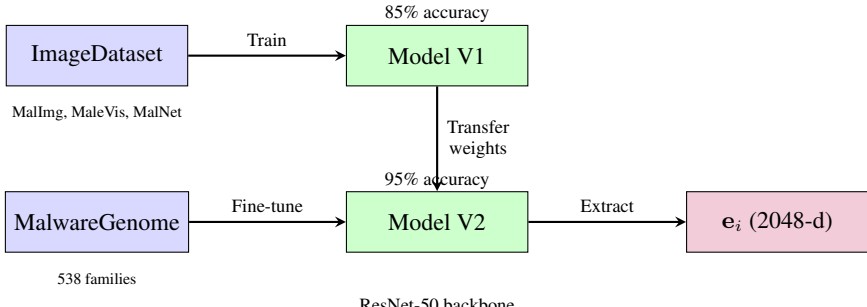

*Figure 17.* Image embedding pipeline. Model V1 (ResNet-50 with ImageNet weights) is trained on public malware image datasets, then fine-tuned as Model V2 on our dataset. The 2048-dimensional penultimate layer activation serves as $\mathbf{e}_i$.

*Table 7.* Hyperparameters for image embedding models (V1 and V2).

| Parameter | Setting |
|---|---|
| Architecture | ResNet-50 |
| Optimizer | Adam |
| Initial learning rate | 0.01 |
| LR scheduler | ReduceLROnPlateau (factor: 0.1, patience: 10) |
| Early stopping patience | 10 epochs |
| Weight decay | 0.0001 |
| Batch size | 64 |
| Train/validation split | 70/30 (stratified) |
| Random seed | 42 |
| Output embedding dimension | 2048 |

## L.2. Embedding Fusion Network

Figure 18 illustrates the fusion architecture. Multi-modal embeddings are concatenated, normalized, and passed through a two-layer network to produce a unified 1000-dimensional representation.

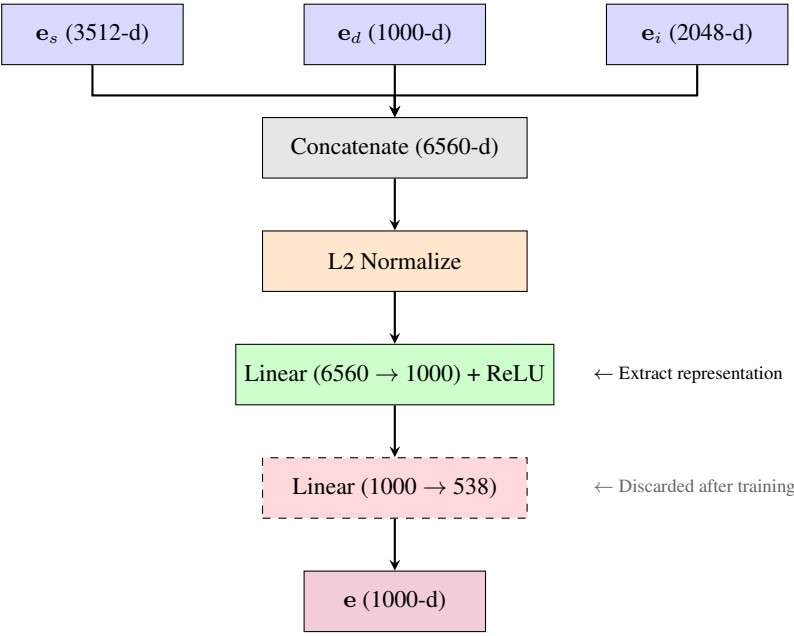

*Figure 18.* Embedding fusion architecture. Multi-modal embeddings are concatenated, normalized, and passed through a two-layer network. After training with cross-entropy loss, the classification head (dashed) is discarded and the 1000-dimensional hidden representation serves as the final embedding.

*Table 8.* Hyperparameters for supervised dimensionality reduction.

| Parameter | Setting |
|---|---|
| Input dimension | 6560 |
| Hidden dimension | 1000 |
| Output dimension | 538 (number of families) |
| Optimizer | Adam |
| Initial learning rate | 0.01 |
| LR scheduler | ReduceLROnPlateau (factor: 0.1, patience: 10) |
| Early stopping patience | 10 epochs |
| L2 regularization | $\lambda = 0.0001$ |
| Batch size | 64 |
| Train/validation split | 70/30 (stratified) |

## L.3. Supervised Fusion Validity

Supervised training on family labels optimizes embeddings for family separability, raising the question of whether phylogenetic algorithms recover genuine evolutionary relationships or merely reconstruct the known taxonomy. Standard train/test validation does not apply: tree topology depends on all samples simultaneously, and no ground-truth tree exists. We therefore validate using VirusTotal timestamps, which are *independent* of family labels. If embeddings encoded only family membership, we would expect near-random temporal ordering within families. High temporal consistency thus indicates that embeddings capture evolutionary divergence beyond what supervision provides.

## M. Feature Extraction Details

We analyze three executable formats: PE (Windows), ELF (Linux/Unix), and DOS (legacy systems).

**PE Format.**   PE files begin with a DOS header (MZ signature) followed by the PE header containing machine architecture, section count, and timestamp. The Optional Header specifies entry point, image base, section/file alignment, and memory allocation (stack/heap sizes). Data Directories point to import/export tables, resources, and relocations. Sections include `.text` (code), `.data` (initialized data), `.rdata` (read-only data), and `.rsrc` (resources). We target sections marked with `MEM_EXECUTE` flags for analysis.

**ELF Format.**   ELF files contain an ELF Header specifying file type (`ET_EXEC` for executables, `ET_DYN` for shared objects), machine architecture, and entry point. The `e_ident[EI_CLASS]` field distinguishes 32-bit (`ELFCLASS32`) from 64-bit (`ELFCLASS64`) binaries. The Program Header Table describes memory segments, while the Section Header Table details `.text`, `.data`, `.bss`, and `.dynsym` sections. We identify executable segments via `SEGMENT_FLAGS.X`.

**DOS Format.**   DOS executables use the MZ header containing block counts, relocation entries, header size, memory allocation requirements, and initial stack segment. Code is analyzed from offset $\texttt{initial\_cs} \times 16 + \texttt{initial\_ip}$ with 16-bit disassembly via Capstone  (Quynh, 2014).

*Table 9.* Extracted features across executable formats.

| Feature | Description | Tools |
|---|---|---|
| *Byte-level* | | |
| ByteHistogram | Byte frequency in executable sections (256-d) | LIEF, NumPy |
| ByteEntropyHistogram | Joint byte-entropy distribution (256-d) | LIEF, NumPy |
| Strings | String count, lengths, paths, URLs, patterns | re, NumPy |
| *Structural* | | |
| GeneralFileInfo | File size, import/export counts, flags | LIEF |
| HeaderFileInfo | Timestamps, architecture, versions | LIEF |
| SectionInfo | Section names, sizes, entropy, permissions | LIEF |
| SegmentInfo | Segment types, sizes, addresses (ELF) | LIEF |
| DataDirectories | Directory sizes and addresses (PE) | LIEF |
| *Import/Export* | | |
| ImportsInfo | Imported libraries and functions (hashed) | LIEF |
| ExportsInfo | Exported functions (hashed) | LIEF |
| *Control Flow* | | |
| EntryPoints | Execution start addresses | LIEF, Capstone |
| ExitPoints | Termination functions/interrupts | LIEF, Capstone |
| Opcodes | Instruction mnemonics | Capstone |
| OpcodeOccurrences | Opcode frequency distribution | Capstone |
| InterruptInfo | Interrupt calls and vectors (DOS) | Capstone |
| *Memory Layout* | | |
| ImageSize | Total virtual size | LIEF |
| HeaderSize | Combined header sizes | LIEF |
| MemorySize | Memory allocation requirements | LIEF |
| StackReserveSize | Reserved stack memory (PE) | LIEF |
| StackCommitSize | Committed stack memory (PE) | LIEF |
| StackInfo | Initial SP/SS values (DOS) | struct |
| GNUStackSize | Stack segment size (ELF) | LIEF |
| HeapSize | Heap reserve and commit sizes | LIEF |
| LoaderFlags | Loader behavior flags | LIEF |
| *Complexity* | | |
| BlockEntropy | Min, max, mean, total entropy | Python |
| SectionEntropies | Per-section entropy statistics (ELF) | LIEF |
| KolmogorovComplexity | Compression ratio | zlib |

# N. Visualization

**Further details for Figure 1.** We constructed the full Neighbor-Joining tree from all 103,883 samples and derived the inter-family graph using the methodology described in Appendix H. For Figure 1, we selected 32 representative families (8 per functional category) based on sample count $> 50$ to ensure stable within-family topology. Using ete3, we pruned the full NJ tree to retain only leaves from these 32 families, preserving original branch lengths, then collapsed each family to a single node for visualization.

**Functional Categories.** We assigned families to functional categories based on their primary documented functionality in public threat intelligence sources (e.g., MITRE ATT&CK, vendor reports). For example, Mirai and its variants are documented IoT botnets; Zeus and Emotet are banking trojans/stealers; NjRat and AsyncRAT are remote access trojans; LockBit and Conti are ransomware. These are established categorizations in the security community, not novel classifications. The categories are not mutually exclusive in practice (some malware exhibits multiple behaviors), but we assigned each family to its primary functional role for visualization purposes.

**Further details for the interactive online figure.** Appendix H was used for each family pair, where we computed median distances from each family's leaves to their shared MRCA, created directed edges from earlier-diverging to later-diverging families (edge weight = source family's median distance), then retained only the minimum-weight outgoing edge per node to focus on primary lineages. The resulting graph contains all 538 families as nodes, with edges representing the strongest inferred evolutionary relationships. This is not a subsample of families but rather a simplification of the edge structure to highlight primary lineages while preserving all families.

# O. Cross-Modality Drift Agreement

A natural concern is whether the pseudo-static, dynamic, and image modalities encode conflicting notions of evolutionary rate, which fusion would then blend incoherently. To test this, we place the modalities on a common scale by normalizing each family's cross-year embedding drift by its within-year spread, yielding a dimensionless signal-to-noise ratio comparable across modalities. This follows the same principle as Wright's $F_{ST}$ (Wright, 1951) and Cohen's $d$ (Cohen, 1988), which scale between-group effects by within-group variation.

Across 272 families, the normalized drift rates are concordant (Figure 19): families that drift rapidly in one modality tend to drift rapidly in the others. The agreement is strongest between the image and pseudo-static modalities (Spearman $\rho = 0.90$) and remains substantial between pseudo-static and dynamic ($\rho = 0.75$). We find no evidence that the modalities provide conflicting evolutionary signals, which is consistent with the fused embedding achieving 87.1% temporal consistency against independent VirusTotal timestamps. Explicitly modeling per-modality trees is a possible refinement, but in the absence of evidence for cross-modality inconsistency the fused representation is well supported.

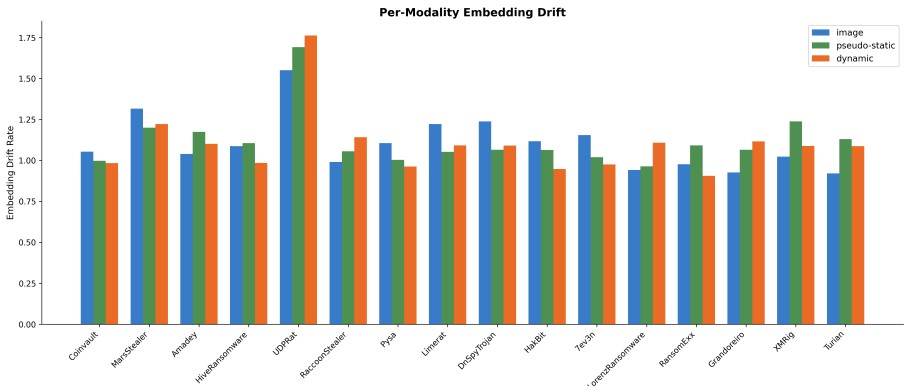

*Figure 19.* Cross-modality drift agreement. Axes are normalized drift rates (cross-year drift scaled by within-year spread) for two modalities. Families that drift rapidly in one modality do so in the others, with Spearman $\rho = 0.90$ between image and pseudo-static and $\rho = 0.75$ between pseudo-static and dynamic across 272 families.

