# OpenReview forum: "MalTree: Tracing Malware Evolution using Embeddings at Scale"
_ICML.cc/2026/Conference — ICML 2026 regular_

### Official Review · Reviewer_ZRWq · 2026-03-05

**Soundness:** 4
**Presentation:** 4
**Significance:** 4
**Originality:** 4
**Overall Recommendation:** 5
**Confidence:** 3

**Summary:**

Malware detection systems face performance degradation as threats constantly evolve, necessitating proactive, lineage-aware analysis. The authors propose MalTree, a framework applying bioinformatics-inspired phylogenetic techniques at scale to automatically model malware evolution. By extracting and fusing pseudo-static, dynamic, and image-based embeddings from a large collection of malware samples, the system constructs distance matrices and generates phylogenetic trees using Neighbor-Joining and UPGMA algorithms. To address the lack of ground-truth divergence data, the authors introduce a temporal validation method using VirusTotal timestamps, achieving an 87.1% temporal consistency score and successfully mapping documented malware lineages like the Mirai botnet.

**Compliance With Llm Reviewing Policy:**

Affirmed.

**Final Justification:**

The authors have addressed my minor concern, I would like to keep my positive score.

**Key Questions For Authors:**

- Could the authors provide the temporal consistency score before these statistical outliers are removed to establish a true baseline?

**Limitations:**

yes

**Strengths And Weaknesses:**

Strengths:
- This paper tackles a critical bottleneck in cybersecurity: the scalability of phylogenetic tree construction for malware. By shifting from computationally heavy, character-based sequence alignment to distance-based methods operating on fused continuous embeddings, the authors successfully scale the analysis to 100k+ samples across 500+ families.
- The introduction of the temporal validation framework using VirusTotal first-submission timestamps is a novel contribution. It provides a quantifiable metric (temporal consistency) for an unsupervised tree-building process where ground-truth evolutionary data is historically absent.
- The practical implication of MalTree is significant. Accelerating threat intelligence and reverse engineering is crucial, where newly emerged threats can be mapped to the phylogenetic tree and hypothesized about their origins.

Weaknesses:
- The two-stage outlier detection process is potentially exposed to selection bias. By removing samples that are statistically distant from their VirusTotal family median before building the final tree and running the temporal validation, the methodology risks artificially inflating the temporal consistency score. I recommend including an ablation study that reports the temporal consistency score with the outliers left in the tree to provide a transparent baseline.

---

> ### Author Rebuttal · Authors · 2026-03-30
>
> We thank the reviewer for the positive assessment and the specific, actionable suggestion regarding outlier removal. We ran the requested ablation.
>
> **Outlier removal ablation.** The reviewer asks whether removing statistical outliers before temporal validation artificially inflates the consistency score. We computed temporal consistency on both the full unfiltered tree and the cleaned tree after IQR-based outlier removal.
>
> |                        | Temporal Consistency | Leaves        |
> |------------------------|----------------------|---------------|
> | **Before outlier removal** | 87.11%               | 103,883       |
> | **After outlier removal**  | 88.54%               | 98,498        |
> | **Difference**             | +1.43%               | 5,385 removed |
>
> The 87.1% reported in Table 2 was computed on the full tree with all 103,883 samples, prior to any filtering. Outlier removal yields only a 1.43 percentage point improvement. If filtering were artificially inflating the score we would expect a much larger jump. The small improvement is consistent with removing genuinely noisy samples, primarily mislabeled families from incorrect VirusTotal consensus and extreme polymorphic variants, rather than selection bias. We hope the ablation above fully addresses the concern regarding selection bias.

---

> > ### Author Rebuttal · Reviewer_ZRWq · 2026-04-01
> >
> > I appreciate the authors for conducting further analysis on temporal consistency, and I maintain the positive score of 5.

---

### Official Review · Reviewer_Gwzv · 2026-03-12

**Soundness:** 3
**Presentation:** 2
**Significance:** 3
**Originality:** 2
**Overall Recommendation:** 4
**Confidence:** 1

**Summary:**

This paper takes a practical perspective based on software evolution and leverages this idea to improve malware detection. The proposed approach achieves strong performance in experiments.

**Compliance With Llm Reviewing Policy:**

Affirmed.

**Final Justification:**

I appreciate the authors for addressing most of my concerns, and I therefore raise my score to 4.

**Key Questions For Authors:**

Please see weaknesses.

**Limitations:**

Yes

**Strengths And Weaknesses:**

Strengths:

**1.** Applying the idea of software evolution to the malware detection task sounds like a very reasonable approach, as existing methods indeed have certain limitations.

**2.** Tree-based methods align with the evolutionary patterns observed in software evolution.

Weaknesses:

**1.** How does the proposed method model different level of embedding evolution?

**2.** The paper does not consider the inconsistent evolution of embeddings across different modalities.

---

> ### Author Rebuttal · Authors · 2026-03-30
>
> We thank the reviewer for the feedback.
>
> **Concern: How does the proposed method model different levels of embedding evolution?**
>
> We thank you for this interesting question. However, we do not know if we interpret it correctly. Could you please clarify this? If you are asking how the framework handles evolution at different granularities. The answer is that the tree naturally operates at multiple levels without requiring separate modeling for each.
>
> At the finest level, individual samples sit as leaves in the tree. Branch lengths between siblings capture small divergences like a variant being recompiled with a minor change. Our temporal validation (Section 4.2) works at this level, comparing sibling pairs, and achieves 87.1% consistency with VirusTotal timestamps.
>
> At the family level, the inter-family analysis (Appendix E) aggregates by computing median distances from each family's leaves to their shared ancestor. This is how we produce the graphs in Figure 6 and Appendix F, where edge weights reflect coarser evolutionary relationships between entire families.
>
> At the broadest level, Figure 1 shows that families with similar functional roles (IoT botnets, stealers, RATs, ransomware) cluster together in the tree without being told to. This emerges naturally from shared code patterns and behavioral strategies across families in the same category.
>
>  **Concern: Cross-modality inconsistency in embedding evolution**
>
> We thank the reviewer for raising this point. We carried out a small experiment to investigate this.
>
> To place rates from different modalities on a common scale, we normalize each family’s cross-year drift by its within-year spread, producing a dimensionless ratio that is directly comparable across modalities and families. This signal-to-noise normalization follows the same principle as Wright (1951) and Cohen's d (Cohen, 1988), both of which scale between-group effects by within-group variation.
>
> Result: https://imgur.com/a/4f8uQ5p
>
> As shown in the figure, drift rates are consistent across modalities. Families that exhibit high drift in one modality tend to do so in the others as well. For example, UDPRat shows rapid drift across all three modalities, whereas Coinvault and Turian evolve slowly across all three. The modalities do not provide conflicting evolutionary signals. They agree on which families evolve quickly and which evolve slowly.
>
> Based on this analysis, we find no evidence that cross-modality inconsistency is a concern in our data. The modalities capture compatible evolutionary signals, which likely explains why the fused embedding achieves 87.1 percent temporal consistency when compared against independent VirusTotal timestamps. While explicitly modeling per-modality trees is an interesting direction for future work, it is one of many possible refinements. In the absence of evidence that such inconsistencies materially affect the inferred evolutionary structure, we believe the current approach is well supported.
>
> **Sources**:
>
> Wright, S. (1951). The genetical structure of populations. Annals of Eugenics, 15:323–354. doi:10.1111/j.1469-1809.1949.tb02451.x
>
> Cohen, J. (1988). Statistical Power Analysis for the Behavioral Sciences (2nd ed.). Lawrence Erlbaum Associates.

---

> > ### Author Rebuttal · Reviewer_Gwzv · 2026-04-01
> >
> > I appreciate the author for resolving most of my concerns, so I’m giving a positive rating.

---

### Official Review · Reviewer_bRsG · 2026-03-12

**Soundness:** 3
**Presentation:** 3
**Significance:** 3
**Originality:** 3
**Overall Recommendation:** 4
**Confidence:** 5

**Summary:**

The paper proposes MalTree, a framework that constructs large-scale phylogenetic trees of malware using multimodal embeddings derived from static structure, dynamic behavior, and image representations of binaries. Pairwise embedding distances are used to build evolutionary trees with distance-based phylogenetic algorithms (UPGMA and Neighbor-Joining), and the resulting trees are evaluated through a temporal consistency metric based on VirusTotal submission timestamps. The study analyzes over 100k malware samples across 538 families and demonstrates that the inferred trees show relatively high temporal consistency and recover several known lineage relationships such as variants of the Mirai botnet. While the large-scale analysis and the evolutionary perspective are interesting, the current work mainly relies on embedding similarity and does not provide code-level evidence of actual inheritance or mutation patterns between malware families. A detailed reverse-engineering case study demonstrating concrete code reuse or evolution along the inferred tree would significantly strengthen the claim that the method captures real malware evolution.

**Compliance With Llm Reviewing Policy:**

Affirmed.

**Final Justification:**

The rebuttal partially addresses my concerns and makes the claims more plausible, and while I still find the analysis not fully solid at the code level, the overall direction is interesting and potentially impactful, so I revise my score to a weak accept.

**Key Questions For Authors:**

Can the authors provide code-level validation cases?

The current case studies mainly rely on known relationships reported in threat intelligence. It would significantly strengthen the paper if the authors could provide reverse-engineering or code-level analyses for some of the inferred lineages, for example by showing shared functions, reused modules, or control-flow similarities between malware families. Such evidence would help demonstrate that the inferred relationships correspond to actual code inheritance rather than only embedding similarity.

**Limitations:**

The main limitation of this work is that the inferred phylogenetic structure is built entirely from embedding distances, so the method currently provides stronger evidence for representation-level similarity than for true code-level evolutionary lineage. While the temporal validation suggests that the recovered tree is correlated with approximate emergence order, it does not directly verify that the inferred parent–child relationships correspond to actual code inheritance, module reuse, or concrete mutation patterns between malware families. In addition, the temporal validation relies on VirusTotal first-submission timestamps as a proxy for emergence time, which may introduce noise because submission time does not necessarily match the true creation or first-circulation time of a malware sample. Finally, malware evolution in practice often involves code borrowing from multiple sources, shared builders, and copy–paste reuse, which may be better modeled as a graph or network rather than a strictly tree-structured process.

**Strengths And Weaknesses:**

Strengths

1. Interesting perspective. The paper studies malware ecosystems from an evolutionary viewpoint rather than treating malware detection purely as a classification problem. This perspective is potentially valuable and provides an alternative way to analyze relationships among malware families.

2. Large-scale analysis. The work evaluates the approach on more than 100k malware samples across over 500 families, which is a relatively large scale compared to many prior studies and demonstrates that the framework can operate at scale.

3. Clear and structured pipeline. The proposed framework integrates multiple modalities (pseudo-static features, dynamic behavioral traces, and image representations of binaries), combines them into embeddings, and constructs phylogenetic trees using established algorithms such as UPGMA and Neighbor-Joining.

4. Initial case study examples. The Mirai-related analysis shows that the inferred tree structure can recover several known relationships reported in prior threat intelligence reports.

Weaknesses

1. Relies primarily on embedding similarity. The inferred phylogenetic tree is constructed entirely from embedding distances, which effectively makes the method closer to large-scale similarity clustering rather than demonstrating true evolutionary relationships.

2. Lack of code-level validation. The paper does not provide reverse-engineering evidence (e.g., decompilation, function-level similarity, or shared code modules) to support that the inferred relationships correspond to actual code inheritance or malware evolution.

3. Limited depth in case studies. The case study mainly confirms relationships that are already known from threat intelligence rather than demonstrating new lineage discoveries supported by deeper technical analysis.

4. Temporal validation assumptions may be noisy. The use of VirusTotal submission timestamps as a proxy for malware emergence time may introduce noise, since submission time does not necessarily reflect the true time when malware was created or first circulated.

5. Limited analysis of evolutionary mechanisms. The paper would be significantly stronger if the inferred relationships were supported by code-level analyses (e.g., shared functions, reused modules, or control-flow similarities) demonstrating concrete evolutionary patterns between malware families.

---

> ### Author Rebuttal · Authors · 2026-03-30
>
> **Concern: Lack of code-level validation**
>
> First we discuss code-level validation that was already present in the paper, and in addition, we provide extra quantative results on code-level validation.
>
> This paper already includes code-level evidence and multiple case studies:
> - Section 5.5: Mirai
> - Appendix F1: Smokeloader
> - F2: Hub-Family effects
> - F3: Conti/Wizard Spider
>
> These results provide new perspectives that traditional reverse engineering does not easily surface. MalTree produces these insights automatically in hours rather than the months typically required for manual reverse engineering.
>
> Now we provide more quantitative results. Consider Mirai. Our pipeline extracts import/export tables (Appendix J, Table 6). Using these features, we compute Jaccard similarity. See below.
>
> ||Mirai|Okiru|Bashlite|Gafgyt|MooBot|
> |-----------|-------|-------|----------|--------|--------|
> |**Mirai**||0.82|0.5|0.07|0.04|
> |**Okiru**|0.82||0.6|0.06|0.05|
> |**Bashlite**|0.58|0.58||0.17|0.04|
> |**Gafgyt**|0.07|0.06|0.17||0.05|
> |**MooBot**|0.04|0.05|0.04|0.05||
>
> Table R1: Jaccard similarity of combined import/export symbol sets.
>
> Compare this with our tree (Section 5.5,  Figure 6 and https://eloquent-bunny-f06d56.netlify.app/). We now examine the code-level features underlying these relationships.
>
> Mirai & Okiru show high overlap (0.82), sharing 44 out of 49/51 unique symbols spanning core IoT botnet functionality: networking, process control , memory management , and file I/O. The small set of Okiru-specific functions (atoi, getdtablesize, getuid, inet_ntoa, sysconf) aligns with expected ARC architecture adaptations (MalwareMustDie, 2018). The tree places them as parent-child (w=10.0), and 82% symbol overlap confirms code reuse. Bashlite has moderate overlap (0.579), consistent Akamai's (2016), and receives the lowest edge weight (w=9.6).
>
> Gafgyt & MooBot have low similarity, reflecting compilation differences. Gafgyt is statically linked with 1377 exported symbols and just 3 imports. MooBot uses a different C runtime (nptl/\_\_cxa vs. uClibc). However, MooBot's export table contains Mirai-specific function names such as attack_method_udpgeneric, attack_udp_ovhhex, setup_connection, resolve_cnc_addr, and anti_gdb_entry, providing direct evidence of code inheritance that import-table comparison alone fails to capture.
>
> **Concern: Relies primarily on embedding similarity / closer to clustering**
>
> We respectfully note that characterizing our approach as clustering overlooks key distinctions between phylogenetic trees and clustering.
>
> Neighbor-Joining (NJ) is not a clustering algorithm; it is a phylogenetic inference method that produces full tree topology. Clustering methods (e.g. k-means) do not infer ancestral nodes, do not yield branch lengths with an evolutionary interpretation, and do not provide tree-recovery guarantees. In contrast, NJ does. Saitou and Nei (1987) showed that it recovers the correct evolutionary topology when distances are additive (see also Atteson (1999)).
>
> Note that, if this were only clustering, our approach should not be able to achieve a high temporal accuracy.
>
> **Concern: Temporal validation assumptions may be noisy**
>
> Indeed VirusTotal timestamps are imperfect (see Section 6). We chose first-submission over file creation dates because the latter proved unreliable (Section 4.2). We also specifically validate on shallow divergences (immediate siblings, Appendix D.3) where estimation error is smallest. As such, we believe we have already mitigated this weakness as much as possible.
>
> **Concern: Better modeled as a graph or network**
>
> We agree and discuss this in Section 6, but network inference remains computationally intractable at our scale. SOTA methods such as SNaQ (Solís-Lemus and Ané, 2016) struggle beyond a few dozen taxa (Hejase et al., 2016), and even recent divide-and-conquer extensions handle at most 81 taxa (Zhu et al., 2019). The underlying problems are NP-hard (Berry et al., 2020). Our dataset contains 103,883 samples. We view MalTree as a foundation demonstrating that evolutionary analysis is feasible and informative at this scale, with network-based extensions as a non-trivial extension for future work.
>
> We hope this addresses the reviewer's concerns and kindly ask the reviewer to reconsider the overall recommendation in light of these new results.
>
> **Sources**:
>
> MalwareMustDie (2018). Okiru: The first Mirai variant targeting ARC processors.
>
> Solís-Lemus & Ané (2016). Inferring phylogenetic networks with maximum pseudolikelihood. PLoS Genetics.
>
> Hejase et al. (2016). A scalability study of phylogenetic network inference methods. BMC Bioinformatics.
>
> Zhu et al. (2019). A divide-and-conquer method for scalable phylogenetic network inference. Bioinformatics.
>
> Berry et al. (2020). Scanning phylogenetic networks is NP-hard. SOFSEM 2020.
>
> Saitou & Nei (1987). The neighbor-joining method. Mol. Biol. Evol.
>
> Atteson (1999). The performance of neighbor-joining methods. Algorithmica.

---

> > ### Author Rebuttal · Reviewer_bRsG · 2026-04-06
> >
> > The rebuttal partially addresses my concerns and makes the claims more plausible, and while I still find the analysis not fully solid at the code level, the overall direction is interesting and potentially impactful, so I revise my score to a weak accept.

---

### Decision · Program_Chairs · 2026-04-30

**Decision:**

Accept (regular)

**Comment:**

There are literature deficiencies in the work that should be resolved in camera ready. Malware analysis is a complex problem that spans a wide array of skills and knowledge, and I as your AC didn't have time to read and flag these concerns to the author in time.

First the use of malware "images" has been known as flawed for some time https://cdn.aaai.org/ocs/ws/ws0432/16422-75958-1-PB.pdf and the use of them successfully implies data deficiencies in the sources used and/or labeling process. Indeed a simple majority labeling approach is used, when better methods are available e.g. https://link.springer.com/chapter/10.1007/978-3-319-45719-2_11 and the improved https://dl.acm.org/doi/abs/10.1145/3701716.3715212 . Others have also looked at source code based validation https://arxiv.org/pdf/2512.00741v1

While these are not trivial issues, they are not totally disqualifying of the work. Proper citation and acknowledgment in the limitations would be appropriate and tempting some of the language in the article with proper deference for the difficulty of malware detection work (e.g., the TLDR of "building the foundation for shifting malware defense from reactive to proactive." is overly strong) would be sufficient to make the paper acceptable.

Indeed it is not fair for myself as the AC to bring up factors that were not part of the review discussion, but the AC has more familiarity with this area and it is hard to balance the details across so many papers. As such, I don't think rejection is warented.

Thus, I recommend the paper for weak acceptance into the proceedings of ICML.